# Information-Theoretic Safe Exploration with Gaussian Processes

**Alessandro G. Bottero**[1,2]**, Carlos E. Luis**[1,2]**, Julia Vinogradska**[1]**, Felix Berkenkamp**[1]**, Jan Peters**[2]
[1]Bosch Center for Artificial Intelligence, Germany
[2]Technische Universität Darmstadt, Germany
`AlessandroGiacomo.Bottero@de.bosch.com`

## Abstract

We consider a sequential decision making task where we are not allowed to evaluate parameters that violate an *a priori* unknown (safety) constraint. A common approach is to place a Gaussian process prior on the unknown constraint and allow evaluations only in regions that are safe with high probability. Most current methods rely on a discretization of the domain and cannot be directly extended to the continuous case. Moreover, the way in which they exploit regularity assumptions about the constraint introduces an additional critical hyperparameter. In this paper, we propose an information-theoretic safe exploration criterion that directly exploits the GP posterior to identify the most informative safe parameters to evaluate. Our approach is naturally applicable to continuous domains and does not require additional hyperparameters. We theoretically analyze the method and show that we do not violate the safety constraint with high probability and that we explore by learning about the constraint up to arbitrary precision. Empirical evaluations demonstrate improved data-efficiency and scalability.

## 1 Introduction

In sequential decision making problems, we iteratively select parameters in order to optimize a given performance criterion. However, real-world applications such as robotics (Berkenkamp et al., 2021), mechanical systems (Schillinger et al., 2017) or medicine (Sui et al., 2015) are often subject to additional safety constraints that we cannot violate during the exploration process (Dulac-Arnold et al., 2019). Since it is *a priori* unknown which parameters lead to constraint violations, we need to actively and carefully learn about the constraints without violating them. That is, we need to learn about the safety of parameters by only evaluating parameters that are currently known to be safe.

Existing methods by Schreiter et al. (2015); Sui et al. (2015) tackle this problem by placing a Gaussian process (GP) prior over the constraint and only evaluate parameters that do not violate the constraint with high probability. To learn about the safety of parameters, they evaluate the parameter with the largest posterior variance. This process is made more efficient by SAFEOPT, which restricts its safe set expansion exploration component to parameters that are close to the boundary of the current set of safe parameters (Sui et al., 2015) at the cost of an additional tuning hyperparameter (Lipschitz constant). However, uncertainty about the constraint is only a proxy objective that only indirectly learns about the safety of parameters. Consequently, data-efficiency could be improved with an exploration criterion that directly maximizes the information gained about the safety of parameters.

**Our contribution** In this paper, we propose Information-Theoretic Safe Exploration (ISE), a safe exploration algorithm that *directly* exploits the information gain about the safety of parameters in order to expand the region of the parameter space that we can classify as safe with high confidence. By directly optimizing for safe information gain, ISE is more data-efficient than existing approaches without manually restricting evaluated parameters to be on the boundary of the safe set, particularly

36th Conference on Neural Information Processing Systems (NeurIPS 2022).

in scenarios where the posterior variance alone is not enough to identify good evaluation candidates, as in the case of heteroskedastic observation noise. This exploration criterion also means that we do not require additional hyperparameters beyond the GP posterior and that ISE is directly applicable to continuous domains. We theoretically analyze our method and prove that it learns about the safety of reachable parameters to arbitrary precision.

**Related work**   Information-based selection criteria with Gaussian processes models are successfully used in the context of unconstrained Bayesian optimization (BO, Shahriari et al. (2016); Bubeck and Cesa-Bianchi (2012)), where the goal is to find the parameters that maximize an *a priori* unknown function. Hennig and Schuler (2012); Henrández-Lobato et al. (2014); Wang and Jegelka (2017) select parameters that provide the most information about the optimal parameters, while Fröhlich et al. (2020) consider the information under noisy parameters. The success of these information-based approaches also relies on the superior data efficiency that they demonstrated. We draw inspiration from these methods when defining an information-based criterion w.r.t. the safety of parameters to guide safe exploration.

In the presence of constraints that the final solution needs to satisfy, but which we can violate during exploration, Gelbart et al. (2014) propose to combine typical BO acquisition functions with the probability of satisfying the constraint. Instead, Gotovos et al. (2013) propose an uncertainty-based criterion that learns about the feasible region of parameters. When we are not allowed to ever evaluate unsafe parameters, safe exploration is a necessary sub-routine of BO algorithms to learn about the safety of parameters. To safely explore, Schreiter et al. (2015) globally learn about the constraint by evaluating the most uncertain parameters. SAFEOPT by Sui et al. (2015) extends this to joint exploration and optimization and makes it more efficient by explicitly restricting safe exploration to the boundary of the safe set. Sui et al. (2018) proposes STAGEOPT, which additionally separates the exploration and optimization phases. Both of these algorithms assume access to a Lipschitz constant to define parameters close to the boundary of the safe set, which is a difficult tuning parameter in practice. These methods have been extended to multiple constraints by Berkenkamp et al. (2021), while Kirschner et al. (2019) scale them to higher dimensions with LINEBO, which explores in low-dimensional sub-spaces. To improve computational costs, Duivenvoorden et al. (2017) suggest a continuous approximation to SAFEOPT without providing exploration guarantees. All of these methods rely on function uncertainty to drive exploration, while we directly maximize the information gained about the safety of parameters.

Safe exploration also arises in the context of Markov decision processes (MDP), (Moldovan and Abbeel, 2012; Hans et al., 2008). In particular, Turchetta et al. (2016, 2019) traverse the MDP to learn about the safety of parameters using methods that, at their core, explore using the same ideas as SAFEOPT and STAGEOPT to select parameters to evaluate. Consequently, our proposed method for safe exploration is also directly applicable to their setting.

## 2   Problem Statement

In this section, we introduce the problem and notation that we use throughout the paper. We are given an unknown and expensive to evaluate safety constraint $f : \mathcal{X} \to \mathbb{R}$ s.t. parameters that satisfy $f(\boldsymbol{x}) \geq 0$ are classified as safe, while others are unsafe. To start exploring safely, we also have access to an initial safe parameter $\boldsymbol{x}_0$ that satisfies the safety constraint, $f(\boldsymbol{x}_0) \geq 0$. We sequentially select safe parameters $\boldsymbol{x}_n \in \mathcal{X}$ where to evaluate $f$ in order to learn about the safety of parameters beyond $\boldsymbol{x}_0$. At each iteration $n$, we obtain a noisy observation of $y_n := f(\boldsymbol{x}_n) + \nu_n$ that is corrupted by additive homoscedastic Gaussian noise $\nu_n \sim \mathcal{N}\left(0, \sigma_\nu^2\right)$. We illustrate the task in Figure 1a, where starting from $\boldsymbol{x}_0$ we aim to safely explore the domain so that we ultimately classify as safe all the safe parameters that are reachable from $\boldsymbol{x}_0$.

As $f$ is unknown and the evaluations $y_n$ are noisy, it is not feasible to select parameters that are safe with certainty and we provide high-probability safety guarantees instead. To this end, we assume that the safety constraint $f$ has bounded norm in the Reproducing Kernel Hilbert Space (RKHS) (Schölkopf and Smola, 2002) $\mathcal{H}_k$ associated to some kernel $k : \mathcal{X} \times \mathcal{X} \to \mathbb{R}$ with $k(\boldsymbol{x}, \boldsymbol{x}') \leq 1$. This assumption allows us to to model $f$ as a Gaussian process (GP) (Srinivas et al., 2010).

**Gaussian Processes**   A GP is a stochastic process specified by a mean function $\mu : \mathcal{X} \to \mathbb{R}$ and a kernel $k$ (Rasmussen and Williams, 2006). It defines a probability distribution over real-valued functions on $\mathcal{X}$, such that any finite collection of function values at parameters $[\boldsymbol{x}_1, \ldots, \boldsymbol{x}_n]$ is

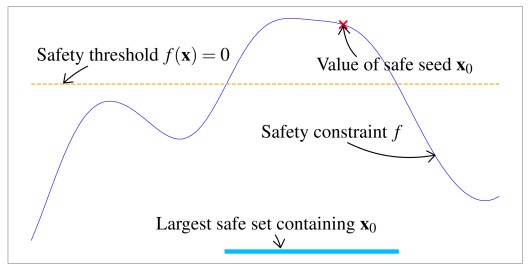

(a) Problem components.

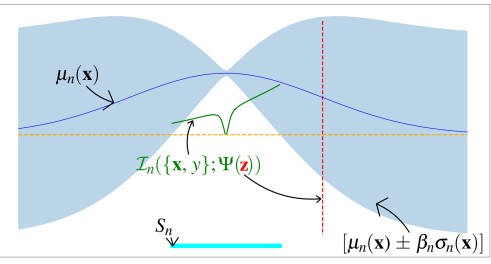

(b) ISE mutual information.

Figure 1: In (a) we illustrate the safe exploration task. Based on the unknown safety constraint $f$, we are only allowed to evaluate safe parameters $\boldsymbol{x}$ with values $f(\boldsymbol{x}) \geq 0$ above the safety threshold (dashed line). Starting from a safe seed $\boldsymbol{x}_0$ a safe exploration strategy needs to discover the largest reachable safe region of the parameter space containing $\boldsymbol{x}_0$. In (b) we show the mutual information $I_n(\{\boldsymbol{x}, y\}; \Psi(\boldsymbol{z}))$ in green for different $\boldsymbol{x}$ inside the safe set and for a fixed $\boldsymbol{z}$ outside (red dashed line). ISE maximizes this mutual information jointly over $\boldsymbol{x}$ and $\boldsymbol{z}$.

distributed as a multivariate normal distribution. The GP prior can then be conditioned on (noisy) function evaluations $\mathcal{D}_n = \{(\boldsymbol{x}_i, y_i)\}_{i=1}^n$. If the noise is Gaussian, then the resulting posterior is also a GP and with posterior mean and variance

$$
\begin{aligned}
\mu_n(\boldsymbol{x}) &= \mu(\boldsymbol{x}) + \boldsymbol{k}(\boldsymbol{x})^\top (\boldsymbol{K} + \boldsymbol{I}\sigma_\nu^2)^{-1}(\boldsymbol{y} - \boldsymbol{\mu}), \\
\sigma_n^2(\boldsymbol{x}) &= k(\boldsymbol{x}, \boldsymbol{x}) - \boldsymbol{k}(\boldsymbol{x})^\top (\boldsymbol{K} + \boldsymbol{I}\sigma_\nu^2)^{-1}\boldsymbol{k}(\boldsymbol{x}),
\end{aligned}
\tag{1}
$$

where $\boldsymbol{\mu} := [\mu(\boldsymbol{x}_1), \ldots \mu(\boldsymbol{x}_n)]$ is the mean vector at parameters $\boldsymbol{x}_i \in \mathcal{D}_n$ and $[\boldsymbol{y}]_i := y(\boldsymbol{x}_i)$ the corresponding vector of observations. We have $[\boldsymbol{k}(\boldsymbol{x})]_i := k(\boldsymbol{x}, \boldsymbol{x}_i)$, the kernel matrix has entries $[\boldsymbol{K}]_{ij} := k(\boldsymbol{x}_i, \boldsymbol{x}_j)$, and $\boldsymbol{I}$ is the identity matrix. In the following, we assume without loss of generality that the prior mean is identically zero: $\mu(\boldsymbol{x}) \equiv 0$.

**Safe set**    Using the previous assumptions, we can construct high-probability confidence intervals on the function values $f(\boldsymbol{x})$. Concretely, for any $\delta > 0$ it is possible to find a sequence of positive numbers $\{\beta_n\}$ such that $f(\boldsymbol{x}) \in [\mu_n(\boldsymbol{x}) \pm \beta_n \sigma_n(\boldsymbol{x})]$ with probability at least $1 - \delta$, jointly for all $\boldsymbol{x} \in \mathcal{X}$ and $n \geq 1$. For a proof and more details see (Chowdhury and Gopalan, 2017). We use these confidence intervals to define a *safe set*

$$
S_n := \{\boldsymbol{x} \in \mathcal{X} : \mu_n(\boldsymbol{x}) - \beta_n \sigma_n(\boldsymbol{x}) \geq 0\} \cup \{\boldsymbol{x}_0\},
\tag{2}
$$

which contains all parameters whose $\beta_n$-lower confidence bound is above the safety threshold and the initial safe parameter $\boldsymbol{x}_0$. Consequently, we know that all parameters in $S_n$ are safe, $f(\boldsymbol{x}) \geq 0$ for all $\boldsymbol{x} \in S_n$, with probability at least $1 - \delta$ jointly over all iterations $n$.

**Safe exploration**    Given the safe set $S_n$, the next question is which parameters in $S_n$ to evaluate in order to efficiently expand it. Most existing safe exploration methods rely on uncertainty sampling over subsets of $S_n$. SAFEOPT-like approaches, for example, use the Lipschitz assumption on $f$ to identify parameters in $S_n$ that could expand the safe set and then select the parameter that has the biggest uncertainty among those. In the next sections, we present and analyze our safe exploration strategy, ISE, that instead uses an information gain measure to identify the parameters that allow us to efficiently learn about the safety of parameters outside of $S_n$.

## 3    Information-Theoretic Safe Exploration

We present Information-Theoretic Safe Exploration (ISE), which guides the safe exploration by using an information-theoretic criterion. Our goal is to design an exploration strategy that directly exploits the properties of GPs to learn about the safety of parameters outside of $S_n$. We draw inspiration from Hennig and Schuler (2012); Wang and Jegelka (2017) who exploit information-theoretic insights to design data-efficient BO acquisition functions for their respective optimization objectives.

**Information gain measure**    In our case, we want to evaluate $f$ at safe parameters that are maximally informative about the safety of other parameters, in particular of those where we are uncertain

**Algorithm 1** Information-Theoretic Safe Exploration

1: **Input:** GP prior $(\mu_0, k, \sigma_\nu)$, Safe seed $\boldsymbol{x}_0$
2: **for** $n = 0, \dots, N$ **do**
3:    $x_{n+1} \leftarrow \arg\max_{\boldsymbol{x} \in S_n} \max_{\boldsymbol{z} \in \mathcal{X}} \hat{I}_n\big(\{\boldsymbol{x}, y\}; \Psi(\boldsymbol{z})\big)$
4:    $y_{n+1} \leftarrow f(\boldsymbol{x}_{n+1}) + \nu$
5:    Update GP posterior with $(\boldsymbol{x}_{n+1}, y_{n+1})$

about whether they are safe or not. To this end, we need a corresponding measure of information gain. We define such a measure using the binary variable $\Psi(\boldsymbol{x}) = \mathbb{I}_{f(\boldsymbol{x}) \geq 0}$, which is equal to one iff $f(\boldsymbol{x}) \geq 0$. Its entropy is given by

$$H_n\big[\Psi(\boldsymbol{z})\big] = -p_n^-(\boldsymbol{z}) \ln\big(p_n^-(\boldsymbol{z})\big) - \big(1 - p_n^-(\boldsymbol{z})\big) \ln\big(1 - p_n^-(\boldsymbol{z})\big) \tag{3}$$

where $p_n^-(\boldsymbol{z})$ is the probability of $\boldsymbol{z}$ being unsafe: $p_n^-(\boldsymbol{z}) = \frac{1}{2} + \frac{1}{2} \operatorname{erf}\left(-\frac{1}{\sqrt{2}} \frac{\mu_n(\boldsymbol{z})}{\sigma_n(\boldsymbol{z})}\right)$. The random variable $\Psi(\boldsymbol{z})$ has high-entropy when we are uncertain whether a parameter is safe or not; that is, its entropy decreases monotonically as $|\mu_n(\boldsymbol{z})|$ increases and the GP posterior moves away from the safety threshold. It also decreases monotonically as $\sigma_n(\boldsymbol{z})$ decreases and we become more certain about the constraint. This behavior also implies that the entropy goes to zero as the confidence about the safety of $\boldsymbol{z}$ increases, as desired.

Given our definition of $\Psi$, we consider the mutual information $I\big(\{\boldsymbol{x}, y\}; \Psi(\boldsymbol{z})\big)$ between an observation $y$ at a parameter $\boldsymbol{x}$ and the value of $\Psi$ at another parameter $\boldsymbol{z}$. Since $\Psi$ is the indicator function of the safe regions of the parameter space, the quantity $I_n\big(\{\boldsymbol{x}, y\}; \Psi(\boldsymbol{z})\big)$ measures how much information about the safety of $\boldsymbol{z}$ we gain by evaluating the safety constraint $f$ at $\boldsymbol{x}$ at iteration $n$, averaged over all possible observed values $y$. This interpretation follows directly from the definition of mutual information: $I_n\big(\{\boldsymbol{x}, y\}; \Psi(\boldsymbol{z})\big) = H_n\big[\Psi(\boldsymbol{z})\big] - \mathbb{E}_y\Big[H_{n+1}\big[\Psi(\boldsymbol{z})\big|\{\boldsymbol{x}, y\}\big]\Big]$, where $H_n[\Psi(\boldsymbol{z})]$ is the entropy of $\Psi(\boldsymbol{z})$ according to the GP posterior at iteration $n$, while $H_{n+1}\big[\Psi(\boldsymbol{z})\big|\{\boldsymbol{x}, y\}\big]$ is its entropy at iteration $n + 1$, conditioned on a measurement $y$ at $\boldsymbol{x}$ at iteration $n$. Intuitively, $I_n\big(\{\boldsymbol{x}, y\}; \Psi(\boldsymbol{z})\big)$ is negligible whenever the confidence about the safety of $\boldsymbol{z}$ is high or, more generally, whenever an evaluation at $\boldsymbol{x}$ does not have the potential to substantially change our belief about the safety of $\boldsymbol{z}$. The mutual information is large whenever an evaluation at $\boldsymbol{x}$ on average causes the confidence about the safety of $\boldsymbol{z}$ to increase significantly. As an example, in Figure 1 we plot $I_n\big(\{\boldsymbol{x}, y\}; \Psi(\boldsymbol{z})\big)$ as a function of $\boldsymbol{x} \in S_n$ for a specific choice of $\boldsymbol{z}$ and for an RBF kernel. As one would expect, we see that the closer it gets to $\boldsymbol{z}$, the bigger the mutual information becomes, and that it vanishes in the neighborhood of previously evaluated parameters, where the posterior variance is negligible.

To compute $I_n\big(\{\boldsymbol{x}, y\}; \Psi(\boldsymbol{z})\big)$, we need to average (3) conditioned on an evaluation $y$ over all possible values of $y$. However, the resulting integral is intractable given the expression of $H_n[\Psi(\boldsymbol{z})]$ in (3). In order to get a tractable result, we derive a close approximation of (3),

$$H_n\big[\Psi(\boldsymbol{z})\big] \approx \hat{H}_n\big[\Psi(\boldsymbol{z})\big] \doteq \ln(2) \exp\left\{-\frac{1}{\pi \ln(2)} \left(\frac{\mu_n(\boldsymbol{z})}{\sigma_n(\boldsymbol{z})}\right)^2\right\}. \tag{4}$$

The approximation in (4) is obtained by truncating the Taylor expansion of $H_n[\Psi(\boldsymbol{z})]$ at the second order, and it recovers almost exactly its true behavior (see Appendix B for details). Since the posterior mean at $\boldsymbol{z}$ after an evaluation at $\boldsymbol{x}$ depends linearly on $\mu_n(\boldsymbol{x})$, and since the probability density of $y$ depends exponentially on $-\mu_n^2(\boldsymbol{x})$, using (4) reduces the conditional entropy $\mathbb{E}_y\Big[\hat{H}_{n+1}\big[\Psi(\boldsymbol{z})\big|\{\boldsymbol{x}, y\}\big]\Big]$ to a Gaussian integral with the exact solution

$$\mathbb{E}_y\Big[\hat{H}_{n+1}\big[\Psi(\boldsymbol{z})\big|\{\boldsymbol{x}, y\}\big]\Big] =$$
$$\ln(2)\sqrt{\frac{\sigma_\nu^2 + \sigma_n^2(\boldsymbol{x})(1 - \rho_n^2(\boldsymbol{x}, \boldsymbol{z}))}{\sigma_\nu^2 + \sigma_n^2(\boldsymbol{x})(1 + c_2 \rho_n^2(\boldsymbol{x}, \boldsymbol{z}))}} \exp\left\{-c_1 \frac{\mu_n^2(\boldsymbol{z})}{\sigma_n^2(\boldsymbol{z})} \frac{\sigma_\nu^2 + \sigma_n^2(\boldsymbol{x})}{\sigma_\nu^2 + \sigma_n^2(\boldsymbol{x})(1 + c_2 \rho_n^2(\boldsymbol{x}, \boldsymbol{z}))}\right\}, \tag{5}$$

where $\rho_n(\boldsymbol{x}, \boldsymbol{z})$ is the linear correlation coefficient between $f(\boldsymbol{x})$ and $f(\boldsymbol{z})$, and with $c_1$ and $c_2$ given by $c_1 := 1/\ln(2)\pi$ and $c_2 := 2c_1 - 1$. This result allows us to analytically calculate the approximated

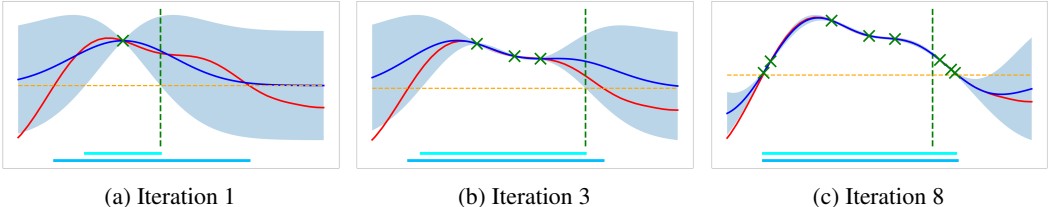

| (a) Iteration 1 | (b) Iteration 3 | (c) Iteration 8 |

Figure 2: Example run of ISE at different iterations. The GP's posterior mean (blue line) and confidence interval $\mu_n \pm \beta_n \sigma_n$ (blue shaded) approximate the true safety constraint $f$ (red line) based on selected data points (green crosses) and, together with the safety threshold at zero (orange line), identify the current safe set $S_n$ (light blue bar, bottom). The vertical green line indicates the location of the next parameter $\boldsymbol{x}_{n+1}$ selected by (6). Initially, ISE evaluates parameters on the boundary of the safe set, but eventually also evaluates inside the safe set if that provides the most information. ISE quickly discover the largest reachable safe set (dark blue bar, bottom).

mutual information $\hat{I}_n\big(\{\boldsymbol{x}, y\}; \Psi(\boldsymbol{z})\big) \doteq \hat{H}_n\big[\Psi(\boldsymbol{z})\big] - \mathbb{E}_y\Big[\hat{H}_{n+1}\big[\Psi(\boldsymbol{z})\big|\{\boldsymbol{x}, y\}\big]\Big]$, which we use to define the ISE acquisition function, and which we analyze theoretically in Section 4.

**ISE acquisition function**   Now that we have defined a way to measure and compute the information gain about the safety of parameters, we can use it to design an exploration strategy that selects the next parameters to evaluate. The natural choice for such selection criterion is to select the parameter that maximizes the information gain; that is, we select $\boldsymbol{x}_{n+1}$ according to

$$\boldsymbol{x}_{n+1} \in \arg\max_{\boldsymbol{x} \in S_n} \max_{\boldsymbol{z} \in \mathcal{X}} \hat{I}_n\big(\{\boldsymbol{x}, y\}; \Psi(\boldsymbol{z})\big), \tag{6}$$

where we jointly optimize over $\boldsymbol{x}$ in the safe set $S_n$ and an unconstrained second parameter $\boldsymbol{z}$. Evaluating $f$ at $\boldsymbol{x}_{n+1}$ according to (6) maximizes the information gained about the safety of some parameter $\boldsymbol{z} \in \mathcal{X}$, so that it allows us to efficiently learn about parameters that are not yet known to be safe. While $\boldsymbol{z}$ can lie in the whole domain, the parameters where we are the most uncertain about the safety constraint lie outside the safe set. By leaving $\boldsymbol{z}$ unconstrained, we show in our theoretical analysis in Section 4 that, once we have learned about the safety of parameters outside the safe set, (6) resorts to learning about the constraint function also inside $S_n$. An overview of ISE can be found in Algorithm 1 and we show an example run of a one-dimensional illustration of the algorithm in Figure 2.

## 4   Theoretical Results

In this section, we study the expression for $\hat{I}_n\big(\{\boldsymbol{x}, y\}; \Psi(\boldsymbol{z})\big)$ obtained using (4) and (5) and analyze the properties of the ISE exploration criterion (6). By construction of $S_n$ (2) and the assumptions on $f$ in Section 2, we know that any parameter selected according to (6) is safe with high probability, see Appendix A for details. To show that we also learn about the safe set, we first need to define what it means to successfully explore starting from $\boldsymbol{x}_0$. The main challenge is that it is difficult to analyze how a GP generalizes based on noisy observations, so that it is difficult to define a notion of convergence that is not dependent on the specific run. SAFEOPT avoids this issue by expanding the safe set not based on the GP, but only using the Lipschitz constant $L$. Contrary to their approach, we depend on the GP to generalize from the safe set. In this case, the natural notion of convergence is provided by the the posterior variance. In particular, we say that at iteration $n$ we have explored the safe set up to $\varepsilon$-accuracy if $\sigma_n^2(\boldsymbol{x}) \leq \varepsilon$ for all parameters $\boldsymbol{x}$ in $S_n$. In the following, we show that ISE asymptotically leads either to $\varepsilon$-accurate exploration of the safe set or to indefinite expansion of the safe set. In future work it will be interesting to further investigate the notion of generalization and to derive a similar convergence result as those obtained by Sui et al. (2015).

**Theorem 1.** *Assume that $\boldsymbol{x}_{n+1}$ is chosen according to* (6)*, and that there exists $\hat{n}$ such that for all $n \geq \hat{n}$ $S_{n+1} \subseteq S_n$. Moreover, assume that for all $n \geq \hat{n}$, $|\mu_n(\boldsymbol{x})| \leq M$ for some $M > 0$ for all $\boldsymbol{x} \in S_n$. Then, for all $\varepsilon > 0$ there exists $N_\varepsilon$ such that $\sigma_n^2(\boldsymbol{x}) \leq \varepsilon$ for every $\boldsymbol{x} \in S_n$ if $n \geq \hat{n} + N_\varepsilon$.*

*The smallest of such $N_\varepsilon$ is given by*

$$N_\varepsilon = \min\left\{ N \in \mathbb{N} : b^{-1}\left(\frac{C\gamma_N}{N}\right) \leq \varepsilon \right\}, \tag{7}$$

*where* $b(\eta) := \ln(2)\exp\left\{-c_1\frac{M^2}{\eta}\right\}\left[1 - \sqrt{\frac{\sigma_\nu^2}{2c_1\eta+\sigma_\nu^2}}\right]$, $\gamma_N = \max_{D\subset\mathcal{X};|D|=N} I\big(\boldsymbol{f}(D); \boldsymbol{y}(D)\big)$ *is the maximum information capacity of the chosen kernel (Srinivas et al., 2010; Contal et al., 2014), and* $C = \ln(2)/\sigma_\nu^2\ln\big(1 + \sigma_\nu^{-2}\big)$.

*Proof.* See Appendix A. ☐

Theorem 1 tells us that if at some point the set of safe parameters $S_n$ stops expanding, then the posterior variance over the safe set vanishes eventually. The intuition behind Theorem 1 is that if there were a parameter $\boldsymbol{x}$ in the safe set whose posterior mean remained finite and whose posterior variance remained bounded from below, then an evaluation of $f$ at such $\boldsymbol{x}$ would yield a non negligible average information gain about the safety of $\boldsymbol{x}$, so that, since $\boldsymbol{x}$ is in the safe set, at some point ISE will be forced to choose to evaluate $\boldsymbol{x}$, reducing its posterior variance. This result guarantees that, should the safe set stop expanding, ISE will asymptotically explore the safe set up to an arbitrary $\varepsilon$-accuracy. In practice, we observe that ISE first focuses on reducing the uncertainty in areas of the safe set that are most informative about parameters whose classification is still uncertain (e.g. the boundary of the safe set), and only eventually turns to learning about the inside of the safe set. This behavior is what ultimately leads to the posterior variance to decay over the whole $S_n$. Therefore, even if in general it is not always possible to say whether or not the safe set will ever stop expanding, we can read Theorem 1 as an exploration guarantee for ISE, as it rules out the possibility that the proposed acquisition function forever leaves the uncertainty high in areas of the safe set that, if better understood, could lead to an expansion of the safe set.

Theorem 1 requires a bound on the GP posterior mean function, which is always satisfied with high probability based on our assumptions about $f$. Specifically, we have that $|\mu_n(\boldsymbol{x})| \leq 2\beta_n$ with probability of at least $1 - \delta$ for all $n$ (see Appendix A for details). Therefore, it does not represent an additional restrictive assumption for $f$. Finally, we also note that the the constant $N_\varepsilon$ defined by (7) always exists since the function $b$ is monotonically increasing, as long as $\gamma_N$ grows sublinearly in $N$. Srinivas et al. (2010) prove that this is the case for commonly-used kernel and, more generally, it is a prerequisite for data-efficient learning with GP models.

## 5   Discussion and Limitations

ISE drives exploration of the parameter space by selecting the parameters to evaluate according to (6). An alternative but conceptually similar approach to this criterion would be to consider the parameter that yields the biggest information gain *on average* over the domain, i.e., substituting the inner max in (6) with an average over $\mathcal{X}$. The resulting integral, however, is intractable and would require further approximations. Moreover, the parameter found by solving (6) will also yield a high average information gain over the domain, due to the regularity of all involved objects.

Being able to work in a continuous domain, ISE can deal with higher dimensional domains better than algorithms requiring a discrete parameter space. However, as noted in Section 4, finding $\boldsymbol{x}_{n+1}$ as in (6) means to solve a non-convex optimization problem with twice the dimension of the the parameter space, which can also become a computationally challenging problem as the dimension grows. In a high-dimensional setting, we follow LINEBO by Kirschner et al. (2019), which at each iteration selects a random one-dimensional subspace to which it restricts the optimization of the acquisition function.

In Sections 2 and 3, we assumed the observation process to be homoskedastic. However, it needs not to be the case, and the results can be extended to the case of heteroskedastic Gaussian noise. The observation noise at a parameter $\boldsymbol{x}$ explicitly appears in the ISE acquisition function, since it crucially affects the amount of information that we can gain by evaluating the constraint $f$ at $\boldsymbol{x}$. On the contrary, STAGEOPT-like methods do not consider the observation noise in their acquisition functions. As a consequence, ISE can perform significantly better in an heteroskedastic setting, as we also show in Section 6.

Lastly, we reiterate that the theoretical safety guarantees offered by ISE are derived under the assumption that $f$ is a bounded norm element of the RKHS space associated with the GP's kernel. In applications, therefore, the choice of the kernel function becomes even more crucial when safety is an issue. For details on how to construct and choose kernels see (Garnett, 2022). The safety guarantees also depend on the choice of $\beta_n$. Typical expressions for $\beta_n$ include the RKHS norm of the constraint $f$ (Chowdhury and Gopalan, 2017; Fiedler et al., 2021), which is in general difficult to estimate in practice. Because of this, usually in practice a constant value of $\beta_n$ is used instead.

# 6   Experiments

In this section we empirically evaluate ISE. Additional details about the experiments and setup can be found in Appendix C. As commonly done in the literature (see Section 5), we set $\beta_n = 2$ for all experiments. This choice guarantees safety per iteration, rather than jointly for all $n$ and it allows for a less conservative bound than the one needed for the joint guarantees.

**GP samples**    For the first part of the experiments, we evaluate ISE on constraint functions $f$ that we obtain by sampling a GP prior at a finite number of points. This allows us to test ISE under the assumptions of the theory and we compare its performance to that of the exploration part of STAGEOPT (Sui et al., 2018). STAGEOPT is a modified version of SAFEOPT, in which the exploration and optimization parts are performed separately: first the SAFEOPT exploration strategy is used to expand the safe set as much as possible, then the objective function is optimized within the discovered safe set. We further modify the version of STAGEOPT that we use in the experiment by defining the safe set in the same way ISE does, i.e., by means of the GP posterior, as done, for example, also by Berkenkamp et al. (2016). We select 100 samples from a two-dimensional GP with RBF kernel, defined in $[-2.5, 2.5] \times [-2.5, 2.5]$ and run ISE and STAGEOPT for 100 iterations for each sample. As STAGEOPT requires a discretization of the domain, we use this discretization to compare the sample efficiency of the two methods, by computing, at each iteration, what percentage of the discretized domain is classified as safe. Moreover, we also compare with the heuristic acquisition inspired by SAFEOPT proposed by Berkenkamp et al. (2016). This method works exactly as STAGEOPT, with the difference that the set of expanders is computed using directly the GP posterior, rather than the Lipschitz constant. More precisely, a parameter $\boldsymbol{x}$ is considered an expander if observing a value of $\mu_n(\boldsymbol{x}) + \beta_n \sigma_n(\boldsymbol{x})$ at $\boldsymbol{x}$ would enlarge the safe set. For the STAGEOPT run, we use the kernel metric to compute the set of potential expanders, for different values of the Lipschitz constant $L$. From the results shown in Figure 3a, we see not only that ISE performs as well or better than all tested instances of STAGEOPT, but also how the choice of $L$ affects the performance of the latter. This plot makes it also evident how crucial the choice of the Lipschitz constant is for STAGEOPT and SAFEOPT-like algorithms in general. In Table 1, in Appendix C, we report the average percentage of safety violations per run achieved by ISE and STAGEOPT. As expected, we see that the percentage of safety violations is comparable among all algorithms.

To show that for STAGEOPT exploration not only overestimating the Lipschitz constant, but also underestimating it can negatively impact performance, we consider the simple one-dimensional constraint function $f(x) = e^{-x} + 0.05$ and run the safe exploration for multiple values of the Lipschitz constant. This function gets increasingly away from the safety threshold for $x \to -\infty$, while it asymptotically approaches the threshold for $x \to \infty$, so that a good exploration algorithm would, ideally, quickly classify as safe the domain region for $x < 0$ and then keep exploring the boundary of the safe set for $x > 0$. The results plotted in Figure 3b show how both a too high and a too low Lipschitz constant can lead to sub-optimal exploration. In the case of a too small constant, this is because STAGEOPT considers expanders almost all parameters in the domain, leading to additional evaluations in the region for $x < 0$ that are unlikely to cause expansion of the safe set. On the other hand, a too high value of the Lipschitz constant can lead to the set of expanders to be empty as soon as the posterior mean gets close to the safety threshold for $x > 0$.

**OpenAI Gym Control**    After investigating the performance of ISE under the hypothesis of the theory, we apply it to two classic control tasks from the OpenAI Gym framework (Brockman et al., 2016), with the goal of finding the set of parameters of a controller that satisfy some safety constraint. In particular we consider linear controllers for the inverted pendulum and cart pole tasks.

For the inverted pendulum task, the linear controller is given by $u_t = \alpha_1 \theta_t + \alpha_2 \dot{\theta}_t$, where $u_t$ is the control signal at time $t$, while $\theta_t$ and $\dot{\theta}_t$ are, respectively, the angular position and the angular velocity

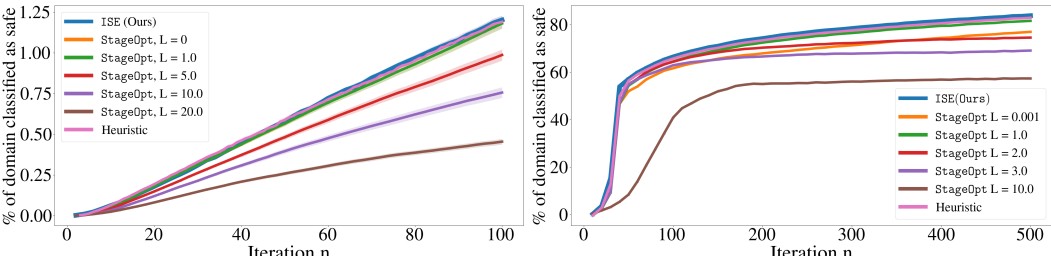

(a) Comparison with STAGEOPT for GP samples.   (b) Comparison in 1D exponential example.

Figure 3: (a) Average expansion of safe set over 100 two-dimensional GP samples. The average percentage of the domain classified as safe is plotted as a function of $n$ with its standard error. The lines for $L = 0$ and $L = 1$ overlap. We can see that ISE obtains an higher sample efficiency than the best instance of STAGEOPT and a comparable one with the heuristic acquisition function proposed by Berkenkamp et al. (2016). The plot also shows that STAGEOPT performance is heavily affected by the choice of $L$. In (b) the average percentage of the domain classified as safe in the one dimensional example for $f(x) = e^{-x} + 0.05$ is plotted as function of th eiteration $n$ with its standard error, and it shows the detrimental effect of over- and underestimating $L$.

of the pendulum. Starting from a position close to the upright equilibrium, the controller's task is the stabilization of the pendulum, subject to a safety constraint on the maximum velocity reached within one episode. For some given initial controller configuration $\boldsymbol{\alpha}_0 := (\alpha_1^0, \alpha_2^0)$, we want to explore the controller's parameter space avoiding configurations that lead the pendulum to swing with a too high velocity. We apply ISE to explore the $\boldsymbol{\alpha}$-space with $\boldsymbol{x}_0 = \boldsymbol{\alpha}_0$ and the safety constraint being the maximum angular velocity reached by the pendulum in an episode of fixed length. In this case, the safety threshold is not at zero, but rather at some finite value $\dot{\theta}_M$, and the safe parameters are those for which the maximum velocity is below $\dot{\theta}_M$. The formalism developed in the previous sections can be easily applied to this scenario if we consider $f(\boldsymbol{\alpha}) = -(\max_t \dot{\theta}_t(\boldsymbol{\alpha}) - \dot{\theta}_M)$. In Figure 4a we show the true safe set for this problem, while in Figures 4b–4d one can see how ISE safely explores the true safe set. These plots show how the ISE acquisition function (6) selects parameters that are close to the current safe set boundary and, hence, most informative about the safety of parameters outside of the safe set. This behavior eventually leads to the full true safe set to be classified as safe by the GP posterior, as Figure 4d shows.

The cart pole task is similar to the inverted pendulum one, but the parameter space has three dimensions. The controller we consider is given by $u_t = \alpha_1 \theta_t + \alpha_2 \dot{\theta}_t + \alpha_3 \dot{s}_t$, where $\theta_t$ and $\dot{\theta}_t$ are, respectively, the angular position and angular velocity of the pole at time $t$, while $\dot{s}_t$ is the cart's velocity. We set the initial state to zero angular and linear velocity and with the pole close to the vertical position, with the controller's goal being to keep the pole stable in the upright position. A combination of the three parameters $\alpha_1$, $\alpha_2$ and $\alpha_3$ is considered safe if the angle of the pole does not exceed a given threshold within the episode. Again, we can easily cast this safety constraint in terms of the formalism developed in the paper: $f(\boldsymbol{\alpha}) = -(\max_t \theta_t(\boldsymbol{\alpha}) - \theta_M)$, where $\theta_M$ is the maximum allowed angle. Figure 5a shows the expansion of the cart pole $\boldsymbol{\alpha}$ space promoted by ISE, compared with STAGEOPT for different values of the Lipschitz constant. Both methods achieve a comparable sample efficiency and both lead to the classification as safe of the full true safe set.

**High dimensional domains**  Many interesting applications have a high dimensional parameter space. While SAFEOPT-like methods are difficult to apply already with dimension $> 3$ due to the discretization of the domain, ISE can perform well also in four or five dimensions. To see this, we apply ISE to the constraint function $f(\boldsymbol{x}) = e^{-\boldsymbol{x}^2} + 2e^{-(\boldsymbol{x}-\boldsymbol{x}_1)^2} + 5e^{-(\boldsymbol{x}-\boldsymbol{x}_2)^2} - 0.2$. Figure 5b shows the ISE performance in dimension 5. We see that ISE is able to promote the expansion of the safe set, leading to an increasingly bigger portion of the true safe set to be classified as safe.

**Heteroskedastic noise domains**  For even higher dimensions, we can follow a similar approach to LINEBO, limiting the optimization of the acquisition function to a randomly selected one-dimensional subspace of the domain. Moreover, as discussed in Section 5, it is also interesting to test ISE in the case of heteroskedastic observation noise, since the noise is a critical quantity for the ISE acquisition function, while it does not affect the selection criterion of STAGEOPT-like methods. Therefore, in

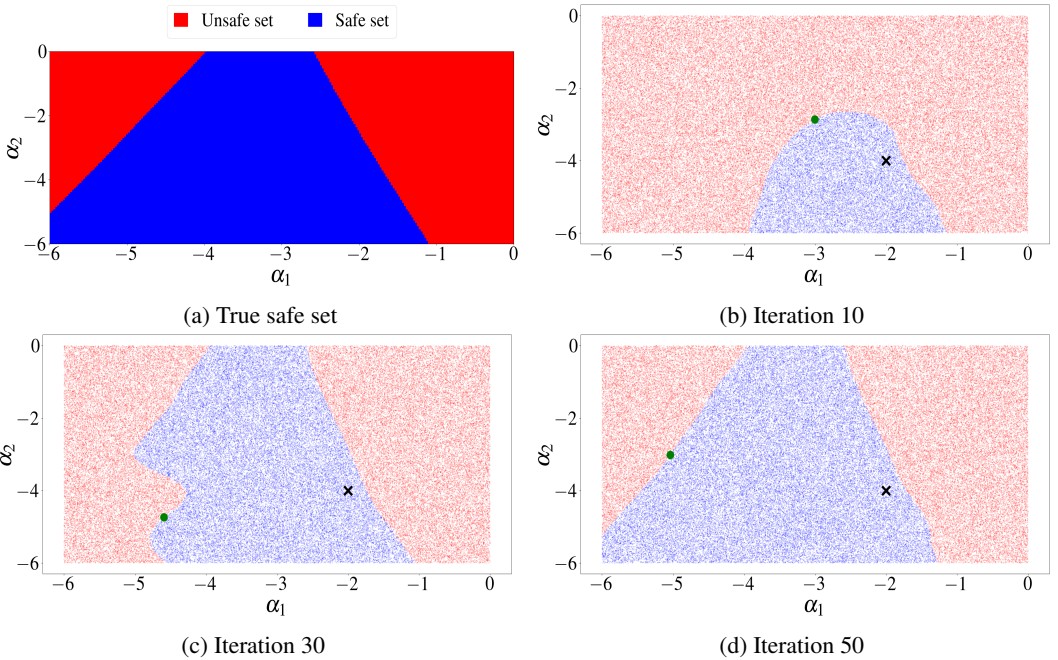

(a) True safe set

(b) Iteration 10

(c) Iteration 30

(d) Iteration 50

Figure 4: Safe exploration of the linear controller's parameter space in the inverted pendulum experiment. In (a) we see the true safe set, while in (b-d) we see the safe set (blue region) as identified by ISE at various iterations. The point marked by the green dot is $\boldsymbol{\alpha}_{n+1}$ as selected by (6), while the black cross is the initial safe seed $\boldsymbol{\alpha}_0$.

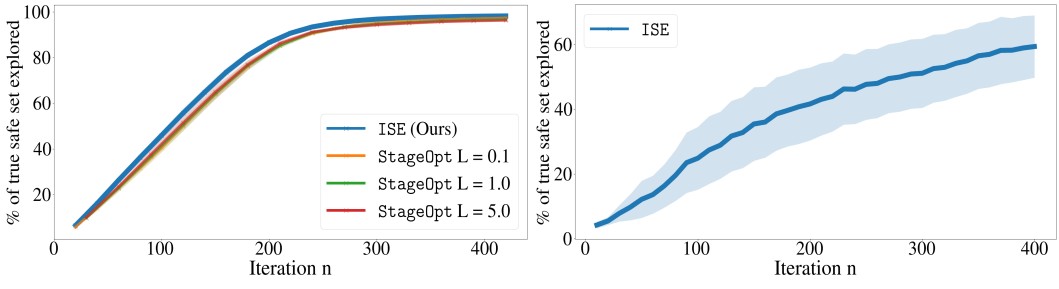

(a) Exploration of the safe set for the cart pole task.

(b) Expansion of the safe set in dimension five.

Figure 5: In (a) we plot the percentage of the true safe set of the cart pole task classified as safe by ISE and STAGEOPT, while in (b) we see the expansion of the five-dimensional safe set promoted by ISE, for the safety constraint $f$ used in the high dimensional experiments.

this experiment we combine a high dimensional problem with heteroskedastic noise. In particular, we apply a LINEBO version of ISE to the constraint function $f(\boldsymbol{x}) = \frac{1}{2}e^{-\boldsymbol{x}^2} + e^{-(\boldsymbol{x} \pm \boldsymbol{x}_1)^2} + 3e^{-(\boldsymbol{x} \pm \boldsymbol{x}_2)^2} + 0.2$ in dimension nine and ten, with the safe seed being the origin. This function has two symmetric global optima at $\pm \boldsymbol{x}_2$ and we set two different noise levels in the two symmetric domain halves containing the optima. To assess the exploration performance, we use the simple regret, defined as the difference between the current safe optimum and the true safe optimum. As the results in Figure 6 show, ISE achieve a greater sample efficiency than the other STAGEOPT-like methods. Namely, for a given number of iterations, by explicitly exploiting knowledge about the observation noise, ISE is able to classify as safe regions of the domain further away from the origin, in which the constraint function assumes its largest values, resulting in a smaller regret. On the other hand, SAFEOPT-like methods only focus on the posterior variance, so that the higher observation noise causes them to remain stuck in a smaller neighborhood of the origin, resulting in bigger regret.

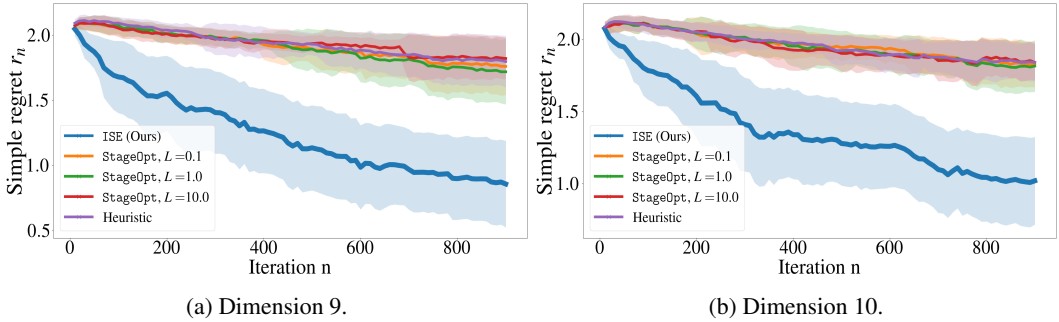

(a) Dimension 9.  (b) Dimension 10.

Figure 6: Example of high dimensional exploration, for $d = 9$ and $d = 10$. After every ten iterations, we perform safe optimization with UCB acquisition within the current safe set and plot the simple regret $r_n$ with respect to the safe optimum. (a) and (a) show, respectively, the average regret over 70 runs in dimension nine and ten, as a function of the number of iterations. We can see that this adapted version of ISE promotes expansion of the safe set, leading to classifying as safe regions where the latent function attains its largest value. The plots also show that ISE achieves a better sample efficiency than both STAGEOPT-like exploration and the STAGEOPT inspired heuristic acquisition.

## 7 Conclusion and Societal Impact

We have introduced Information-Theoretic Safe Exploration (ISE), a novel approach to safely explore a space in a sequential decision task where the safety constraint is *a priori* unknown. ISE efficiently and safely explores by evaluating only parameters that are safe with high probability and by choosing those parameters that yield the greatest information gain about the safety of other parameters. We theoretically analyzed ISE and showed that it leads to arbitrary reduction of the uncertainty in the largest reachable safe set containing the starting parameter. Our experiments support these theoretical results and demonstrate an increased sample efficiency and scalability of ISE compared to SAFEOPT-based approaches.

In many safety sensitive applications the shape of the safety constraints is unknown, so that an important prerequisite for any kind of process is to identify what parameters are safe to evaluate. By providing a principled way to do this, the contributions of this paper allow to deal with safety in a broad range of applications, which can favor the usage of ML approaches also in safety sensitive settings. On the other hand, misuse of the proposed method cannot be prevented in general.

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
