# A Proofs

This appendix contains the proofs of the results found in Section 4. We start by introducing a useful rewriting of the mutual information $\hat{I}_n(\{\boldsymbol{x}, y\}; \Psi(\boldsymbol{z}))$ as given by (4) and (5). We then use this expression to prove some results about $\hat{I}_n(\{\boldsymbol{x}, y\}; \Psi(\boldsymbol{z}))$ needed for the proof of Theorem 1. Finally, we formally show the safety guarantee offered by ISE and the claim that the posterior mean is bounded by $2\beta_n$ with high probability.

**Lemma 1.** *The mutual information $\hat{I}_n(\{\boldsymbol{x}, y\}; \Psi(\boldsymbol{z}))$ as given by (4) and (5) can be rewritten as*

$$\ln(2)\left[\exp\left\{-c_1\frac{\mu_n^2(\boldsymbol{z})}{\sigma_n^2(\boldsymbol{z})}\right\} - \sqrt{\frac{1 - \rho_\nu^2(\boldsymbol{x})\rho_n^2(\boldsymbol{x}, \boldsymbol{z})}{1 + c_2\rho_\nu^2(\boldsymbol{x})\rho_n^2(\boldsymbol{x}, \boldsymbol{z})}}\exp\left\{-c_1\frac{\mu_n^2(\boldsymbol{z})}{\sigma_n^2(\boldsymbol{z})}\frac{1}{1 + c_2\rho_\nu^2(\boldsymbol{x})\rho_n^2(\boldsymbol{x}, \boldsymbol{z})}\right\}\right] \tag{8}$$

*where $c_1 = 1/\ln(2)\pi$ and $c_2 = 2c_1 - 1$, and where $\rho_\nu^2(\boldsymbol{x})$ is given by*

$$\rho_\nu^2(\boldsymbol{x}) := \frac{\sigma_n^2(\boldsymbol{x})}{\sigma_\nu^2 + \sigma_n^2(\boldsymbol{x})} \tag{9}$$

*where the dependency on $n$ has been dropped in the notation for simplicity.*

*Proof.* It suffices to substitute the expression (9) for $\rho_\nu^2(\boldsymbol{x})$ in the second term of (8) to recover (5). The claim follows then directly from (4) and the definition of mutual information. □

**Lemma 2.** *The mutual information $\hat{I}_n(\{\boldsymbol{x}, y\}; \Psi(\boldsymbol{z}))$ as given by (8) is monotonically decreasing in $\mu_n^2(\boldsymbol{z})/\sigma_n^2(\boldsymbol{z}) \,\forall \boldsymbol{x}, \boldsymbol{z} \in \mathcal{X}$.*

*Proof.* First of all, let us simplify notation and define $R^2 := \mu_n^2(\boldsymbol{z})/\sigma_n^2(\boldsymbol{z})$ and $\tilde{\rho}^2 := \rho_\nu^2(\boldsymbol{x})\rho_n^2(\boldsymbol{x}, \boldsymbol{z})$. We then need to show that:

$$\frac{\partial}{\partial R^2}\left[\exp\left\{-c_1 R^2\right\} - \sqrt{\frac{1 - \tilde{\rho}^2}{1 + c_2\tilde{\rho}^2}}\exp\left\{-c_1 R^2\frac{1}{1 + c_2\tilde{\rho}^2}\right\}\right] < 0 \,\forall R, \,\forall \tilde{\rho} \in [0, 1] \tag{10}$$

We then can compute the derivative and ask under which conditions it is non negative. Requiring (10) to be non negative is equivalent to ask that:

$$R^2 \le \frac{1 + c_2\tilde{\rho}^2}{c_1 c_2\tilde{\rho}^2}\left[\ln\left(1 + c_2\tilde{\rho}^2\right) + \frac{1}{2}\ln\left(\frac{1 + c_2\tilde{\rho}^2}{1 - \tilde{\rho}^2}\right)\right] \tag{11}$$

Now, we observe that, since $c_2 \in (-1, 0)$, while $c_1 > 0$ and $\tilde{\rho}^2 \in [0, 1]$, the factor $(1 + c_2\tilde{\rho}^2)/c_1 c_2\tilde{\rho}^2$ is always negative. For what concerns the sum of logarithms in the square brackets, it is strictly positive $\forall \tilde{\rho}^2 \in [0, 1]$, which means that, for (10) to be non negative, we would need $R^2 < 0$, which is impossible, given that $R \in \mathbb{R}$. □

**Lemma 3.** *The mutual information $\hat{I}_n(\{\boldsymbol{x}, y\}; \Psi(\boldsymbol{z}))$ as given by (4) and (5) is monotonically increasing in $\rho_\nu^2(\boldsymbol{x})\rho_n^2(\boldsymbol{x}, \boldsymbol{z}) \,\forall \boldsymbol{x}, \boldsymbol{z} \in \mathcal{X}$.*

*Proof.* As in the proof of Lemma 2, let us define $R^2 := \mu_n^2(\boldsymbol{z})/\sigma_n^2(\boldsymbol{z})$ and $\tilde{\rho}^2 := \rho_\nu^2(\boldsymbol{x})\rho_n^2(\boldsymbol{x}, \boldsymbol{z})$. Analogously to the proof of Lemma 2, we compute the partial derivative of $\hat{I}_n(\{\boldsymbol{x}, y\}; \Psi(\boldsymbol{z}))$ with respect to $\tilde{\rho}^2$ and show that it is strictly positive $\forall \tilde{\rho} \in [0, 1]$ and $\forall R$. The partial derivative is given by:

$$\frac{\partial}{\partial \tilde{\rho}^2}\hat{I}_n(\{\boldsymbol{x}, y\}; \Psi(\boldsymbol{z})) = -\frac{1}{2}\frac{\exp\left\{\frac{c_1 R^2}{|c_2|\tilde{\rho}^2 - 1}\right\}\left[|c_2|^2\tilde{\rho}^2 + |c_2|\left(-2c_1 R^2(\tilde{\rho}^2 - 1) - \tilde{\rho}^2 - 1\right) + 1\right]}{\sqrt{\frac{\tilde{\rho}^2 - 1}{|c_2|\tilde{\rho}^2 - 1}}\left(|c_2|\tilde{\rho}^2 - 1\right)^3} \tag{12}$$

we now have to ask when this derivative is non positive. After remembering that $|c_2| < 1$ and that $\tilde{\rho}^2 \in [0, 1]$, we see that the denominator is always negative; we also have that the exponential term in the numerator is always positive. These two facts, together with the minus sign, imply that, for (12)

to be $\leq 0$, we need the term inside the square brackets to be non positive. This requirement leads to the condition:

$$\tilde{\rho}^2 \geq \frac{c_1 \pi^2 \ln^2(2) R^2 - 2c_1 \pi \ln(2) R^2 + \pi \ln(2)}{\left(\pi \ln(2) - 2\right)\left(c_1 \pi \ln(2) R^2 + 1\right)} \tag{13}$$

where we have used the explicit value of $c_2$: $2c_1 - 1$. Finally, the rhs of (13) is always above 1 for $R^2 \geq 0$, which concludes the proof, since $\tilde{\rho}^2 \in [0, 1]$. $\qquad\square$

**Lemma 4.** *For every value of $\sigma_\nu^2 > 0$, $\rho_\nu^2(\boldsymbol{x})$ as defined in (9) is monotonically increasing in $\sigma_n(\boldsymbol{x})$.*

*Proof.* As for Lemmas 2 and 3, we compute the derivative of $\rho_\nu^2(\boldsymbol{x})$ with respect to $\sigma_n^2(\boldsymbol{x})$ and show that it is strictly positive $\forall \sigma_n^2(\boldsymbol{x})$:

$$\frac{\partial}{\partial \sigma_n^2(\boldsymbol{x})} \rho_\nu^2(\boldsymbol{x}) = \frac{\sigma_\nu^2}{\left(\sigma_n^2(\boldsymbol{x}) + \sigma_\nu^2\right)^2} \tag{14}$$

which is obviously strictly positive $\forall \sigma_\nu^2 > 0$ and $\forall \sigma_n^2(\boldsymbol{x})$. $\qquad\square$

**Lemma 5.** $\forall \boldsymbol{x}, \boldsymbol{z} \in \mathcal{X}$, $\forall n$, $\hat{I}_n\big(\{\boldsymbol{x}, y\}; \Psi(\boldsymbol{z})\big)$ *as given by (4) and (5) is monotonically increasing in* $\hat{H}_n\big[\Psi(\boldsymbol{z})\big]$, $\rho_n^2(\boldsymbol{x}, \boldsymbol{z})$ *and* $\sigma_n^2(\boldsymbol{x})$.

*Proof.* The result follows by combining Lemmas 2–4 with the fact that $\hat{H}_n\big[\Psi(\boldsymbol{z})\big]$ is monotonically decreasing in $\left(\mu_n(\boldsymbol{z})/\sigma_n(\boldsymbol{z})\right)^2$ and that $\rho_n^2 \rho_\nu^2$ is clearly monotonic in $\rho_\nu^2$. $\qquad\square$

**Lemma 6.** *For any finite $\mu_n^2(\boldsymbol{z})/\sigma_n^2(\boldsymbol{z})$, the average entropy variation (8) is non negative for all values of $\rho_n^2(\boldsymbol{x}, \boldsymbol{z})$ and $\rho_\nu^2(\boldsymbol{x})$, and it is zero iff $\rho_\nu^2(\boldsymbol{x})\rho_n^2(\boldsymbol{x}, \boldsymbol{z}) = 0$.*

*Proof.* The result follows immediately from Lemma 3, after noticing that for $\rho_\nu^2(\boldsymbol{x})\rho_n^2(\boldsymbol{x}, \boldsymbol{z}) = 0$ the mutual information (8) is zero and that $\rho_\nu^2(\boldsymbol{x})\rho_n^2(\boldsymbol{x}, \boldsymbol{z})$ is never negative. $\qquad\square$

**Lemma 7.** $\forall n$, $\forall \boldsymbol{x} \in S_n$, $\forall \boldsymbol{z} \in \mathcal{X}$, *it holds that:*

$$\hat{I}_n\big(\{\boldsymbol{x}, y\}; \Psi(\boldsymbol{z})\big) \leq \ln(2) \frac{\sigma_n^2(\boldsymbol{x})}{\sigma_\nu^2} \tag{15}$$

*Proof.* This can be shown directly with the following inequality chain:

$$
\begin{aligned}
\hat{I}_n\big(\{\boldsymbol{x}, y\}; \Psi(\boldsymbol{z})\big) &\leq \ln(2)\left[1 - \sqrt{\frac{1 - \rho_\nu^2(\boldsymbol{x})\rho_n^2(\boldsymbol{x}, \boldsymbol{z})}{1 + c_2 \rho_\nu^2(\boldsymbol{x})\rho_n^2(\boldsymbol{x}, \boldsymbol{z})}}\right] \\
&\overset{c_2 \in (-1, 0)}{\leq} \ln(2)\left(1 - \sqrt{1 - \rho_\nu^2(\boldsymbol{x})\rho_n^2(\boldsymbol{x}, \boldsymbol{z})}\right) \\
&\overset{\rho_n \rho_n^2 \in [0, 1]}{\leq} \ln(2)\left(\rho_\nu^2(\boldsymbol{x})\rho_n^2(\boldsymbol{x}, \boldsymbol{z})\right) \\
&\overset{\rho_n^2 \in [0, 1]}{\leq} \ln(2)\rho_\nu^2(\boldsymbol{x}) \\
&\leq \ln(2)\frac{\sigma_n^2(\boldsymbol{x})}{\sigma_\nu^2}
\end{aligned}
\tag{16}
$$

where the first inequality follows from Lemma 2. $\qquad\square$

**Lemma 8.** $\forall n$, *let $\tilde{\boldsymbol{x}} \in \arg\max_{S_n} \sigma_n^2(\boldsymbol{x})$ and let $M^2 := \max_{S_n} \mu_n^2(\boldsymbol{x})$, and $\tilde{\sigma}^2 := \sigma_n^2(\tilde{\boldsymbol{x}})$, then it holds that:*

$$\hat{I}_n\big(\{\boldsymbol{x}_{n+1}, y_{n+1}\}; \Psi(\boldsymbol{z}_{n+1})\big) \geq b(\tilde{\sigma}^2) \tag{17}$$

*where $b$ is given by*

$$b(\eta) := \ln(2) \exp\left\{-c_1 \frac{M^2}{\eta}\right\}\left[1 - \sqrt{\frac{\sigma_\nu^2}{2c_1\eta + \sigma_\nu^2}}\right]. \tag{18}$$

*Proof.* This Lemma is only non-trivial in case the posterior mean is bounded on $S_n$, otherwise, if we admit $|\mu_n(\boldsymbol{x})| \to \infty$, then we just recover the result that the average information gain is positive.

Now, moving to the proof, as first thing we recall that our algorithm always selects the $\arg\max_{\boldsymbol{x} \in S_n}$ of $\hat{I}_n(\{\boldsymbol{x}, y\}; \Psi(\boldsymbol{z}))$ as next parameter to evaluate, meaning that, by construction, $\forall \boldsymbol{x} \in S_n$ and $\forall \boldsymbol{z} \in \mathcal{X}$, it holds that:

$$\hat{I}_n(\{\boldsymbol{x}_{n+1}, y_{n+1}\}; \Psi(\boldsymbol{z}_{n+1})) \geq \hat{I}_n(\{\boldsymbol{x}, y\}; \Psi(\boldsymbol{z})) \tag{19}$$

This implies, in particular, that $\hat{I}_n(\{\boldsymbol{x}_{n+1}, y_{n+1}\}; \Psi(\boldsymbol{z}_{n+1})) \geq \hat{I}_n(y(\tilde{\boldsymbol{x}}); \Psi(\tilde{\boldsymbol{x}}))$, since, by definition, $\tilde{\boldsymbol{x}} \in S_n$ and is, therefore, always feasible. Writing this condition explicitly, we obtain:

$$
\begin{aligned}
\hat{I}_n(\{\boldsymbol{x}_{n+1}, y_{n+1}\}; \Psi(\boldsymbol{z}_{n+1})) &\geq \hat{I}_n(y(\tilde{\boldsymbol{x}}); \Psi(\tilde{\boldsymbol{x}})) \\
&= \ln(2)\left[\exp\left\{-c_1\frac{\mu_n^2(\tilde{\boldsymbol{x}})}{\tilde{\sigma}^2}\right\} - \sqrt{\frac{1 - \rho_\nu^2(\tilde{\boldsymbol{x}})}{1 + c_2\rho_\nu^2(\tilde{\boldsymbol{x}})}} \exp\left\{-c_1\frac{\mu_n^2(\tilde{\boldsymbol{x}})}{\tilde{\sigma}^2}\frac{1}{1 + c_2\rho_\nu^2(\tilde{\boldsymbol{x}})}\right\}\right] \\
&\geq \ln(2)\exp\left\{-c_1\frac{M^2}{\tilde{\sigma}^2}\right\}\left[1 - \sqrt{\frac{1 - \rho_\nu^2(\tilde{\boldsymbol{x}})}{1 + c_2\rho_\nu^2(\tilde{\boldsymbol{x}})}}\right] \\
&= b(\tilde{\sigma}^2)
\end{aligned}
\tag{20}
$$

where we have used the fact that $c_2 \in (-1, 0)$ and that $\rho_\nu^2(\tilde{\boldsymbol{x}}) \in [0, 1]$. $\square$

**Lemma 9.** *The function $b$ defined in* (18) *is monotonically increasing for positive arguments.*

*Proof.* By looking at the definition of $b$

$$b(\eta) := \ln(2)\exp\left\{-c_1\frac{M^2}{\eta}\right\}\left[1 - \sqrt{\frac{\sigma_\nu^2}{2c_1\eta + \sigma_\nu^2}}\right] \tag{21}$$

we immediately see that both the exponential factor and the term in square brackets are monotonically increasing with the argument $\eta$, if this is positive, so that $b$ is also monotonically increasing with $\eta > 0$. This can also be shown formally by computing the derivative:

$$\frac{1}{\ln(2)}\frac{db(\eta)}{d\eta} = \frac{1}{\eta^2}M^2 c_1 e^{-c_1 M^2/\eta}\left(1 - \sqrt{\frac{\sigma_\nu^2}{2c_1\eta + \sigma_\nu^2}}\right) + \frac{\sigma_\nu^2 c_1 e^{-c_1 M^2/\eta}}{\sqrt{\frac{\sigma_\nu^2}{2c_1\eta + \sigma_\nu^2}}(2c_1\eta + \sigma_\nu^2)^2} \tag{22}$$

and by noticing that, for positive $\eta$, it is always positive, since $c_1 > 0$. $\square$

**Corollary 1.** *From Lemma 9 it follows immediately that, for positive arguments, $b^{-1}$ exists and is also monotonically increasing. In Figure 7 we show some examples of the function $b^{-1}$ for $M = 0.5$ and various values of $\sigma_\nu^2$.*

**Theorem 1.** *Assume that $\boldsymbol{x}_{n+1}$ is chosen according to* (6)*, and that there exists $\hat{n}$ such that for all $n \geq \hat{n}$ $S_{n+1} \subseteq S_n$. Moreover, assume that for all $n \geq \hat{n}$, $|\mu_n(\boldsymbol{x})| \leq M$ for some $M > 0$ for all $\boldsymbol{x} \in S_n$. Then, for all $\varepsilon > 0$ there exists $N_\varepsilon$ such that $\sigma_n^2(\boldsymbol{x}) \leq \varepsilon$ for every $\boldsymbol{x} \in S_n$ if $n \geq \hat{n} + N_\varepsilon$. The smallest of such $N_\varepsilon$ is given by*

$$N_\varepsilon = \min\left\{N \in \mathbb{N} : b^{-1}\left(\frac{C\gamma_N}{N}\right) \leq \varepsilon\right\}, \tag{7}$$

*where $b(\eta) := \ln(2)\exp\left\{-c_1\frac{M^2}{\eta}\right\}\left[1 - \sqrt{\frac{\sigma_\nu^2}{2c_1\eta + \sigma_\nu^2}}\right]$, $\gamma_N = \max_{D \subset \mathcal{X}; |D|=N} I(\boldsymbol{f}(D); \boldsymbol{y}(D))$ is the maximum information capacity of the chosen kernel (Srinivas et al., 2010; Contal et al., 2014), and $C = \ln(2)/\sigma_\nu^2\ln(1 + \sigma_\nu^{-2})$.*

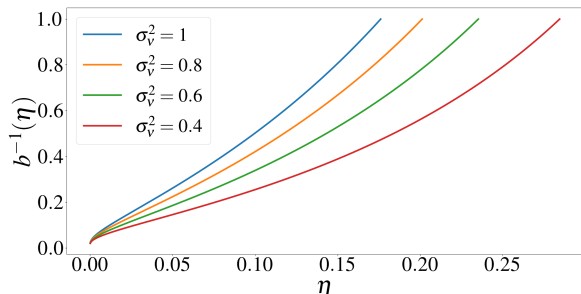

Figure 7: Example plots of the function $b^{-1}$ introduced in Lemma 8 for $M = 0.5$ and different values of $\sigma_\nu^2$.

*Proof.* In the following $n$ will always be intended $\geq \hat{n}$, where $\hat{n}$ is the one given by the hypothesis. Let us also define again $\tilde{\sigma}_n^2 := \max_{S_n} \sigma_n^2(\boldsymbol{x})$. Finally, let us fix $\varepsilon > 0$.

Combining Lemmas 7 and 8, we obtain:

$$b(\tilde{\sigma}_n^2) \leq \hat{I}_n\big(\{\boldsymbol{x}_{n+1}, y_{n+1}\}; \Psi(\boldsymbol{z}_{n+1})\big) \leq \ln(2)\frac{\sigma_n^2(\boldsymbol{x}_{n+1})}{\sigma_\nu^2}$$

$$\implies b(\tilde{\sigma}_n^2) \leq \ln(2)\frac{\sigma_n^2(\boldsymbol{x}_{n+1})}{\sigma_\nu^2} \tag{23}$$

we can now exploit the monotonicity of $b$ (Lemma 9) and the fact that $\tilde{\sigma}_n^2$ is not increasing if the safe set does not expand to conclude that:

$$b(\tilde{\sigma}_n^2) \leq b(\tilde{\sigma}_m^2) \leq \ln(2)\frac{\sigma_m^2(x_{m+1})}{\sigma_\nu^2} \qquad \forall n \geq m \geq \hat{n} \tag{24}$$

we can then use this to write:

$$(n - \hat{n})b(\tilde{\sigma}_n^2) = \sum_{i=\hat{n}}^{n} b(\tilde{\sigma}_n^2) \leq$$

$$\ln(2)\sum_{i=\hat{n}}^{n} \sigma_\nu^{-2}\sigma_i^2(\boldsymbol{x}_{i+1}) \leq \tag{25}$$

$$\frac{\ln(2)}{\sigma_\nu^2 \ln\big(1 + \sigma_\nu^{-2}\big)} \sum_{i=\hat{n}}^{n} \ln\Big(1 + \sigma_\nu^{-2}\sigma_i^2(\boldsymbol{x}_{i+1})\Big) \leq$$

$$C\gamma_{n-\hat{n}}$$

where $\gamma_{n-\hat{n}}$ is the maximum information capacity and $C = \frac{\ln(2)}{\sigma_\nu^2 \ln(1+\sigma_\nu^{-2})}$. The second last passage follows from the fact that $x \leq \ln(1 + x)\sigma_\nu^{-2}/\ln\big(1 + \sigma_\nu^2\big)$ for $x \in [0, \sigma_\nu^{-2}]$ together with the fact that $\sigma_\nu^{-2}\sigma_i^2(\boldsymbol{x}_{i+1}) \leq \sigma_\nu^{-2}k(\boldsymbol{x}_{i+1}, \boldsymbol{x}_{i+1}) \leq \sigma_\nu^{-2}$. Finally, the last passage uses the representation of the mutual information $I(\{y_n\}; \{f(\boldsymbol{x}_n)\})$ presented by Srinivas et al. (2010).

Using (25), we can show that the minimum $N_\varepsilon$ satisfying the claim of the theorem is the one given by (7):

$$N_\varepsilon = \min\left\{N \in \mathbb{N} : b^{-1}\left(\frac{C\gamma_N}{N}\right) \leq \varepsilon\right\} \tag{26}$$

and we are now able to conclude that, as long as the information capacity grows sub-linearly in $N$, the set on the r.h.s. of (26) is not empty $\forall \varepsilon > 0$. This is guaranteed by the fact that $b^{-1}$ is monotonically increasing, since so is its inverse $b$. To check that this $N_\varepsilon$ indeed satisfies the claim, one just has to apply $b^{-1}$ on both initial and final state of (25) and then substitute $\hat{n} + N_\varepsilon$ in the place of $n$; the rest follows from the fact that the maximum variance is non increasing on $S_n$ as long as the safe set does not expand. □

**Corollary 2.** *Under the hypothesis of Theorem 1, $\forall \varepsilon > 0 \, \exists N_\varepsilon \in [0, \infty)$ s.t. $\hat{I}_n\big(\{\boldsymbol{x}, y\}; \Psi(\boldsymbol{z})\big) \leq \varepsilon$ $\forall n \geq \hat{n} + N_\varepsilon$.*

*Proof.* This follows directly from Theorem 1 and from the fact that $\hat{I}_n\big(\{\boldsymbol{x}, y\}; \Psi(\boldsymbol{z})\big)$ is upper bounded by a monotonic function of the posterior variance (Lemma 7). $\qquad\square$

Moving on to the safety guarantees, in Section 4 we claimed that any parameter selected according to (6) is safe with high probability. The following result makes this statement precise.

**Lemma 10.** *Let $f : \mathcal{X} \to \mathbb{R}$ have bounded norm in the Reproducing Kernel Hilbert Space $\mathcal{H}_k$ associated to some kernel $k : \mathcal{X} \times \mathcal{X} \to \mathbb{R}$ with $k(\boldsymbol{x}, \boldsymbol{x}') \leq 1$, and let $S_n$ be the corresponding safe set as defined in (2), with $\boldsymbol{x}_0$ such that $f(\boldsymbol{x}_0) > 0$. Then, if $\boldsymbol{x}_{n+1}$ is selected according to (6), it follows that $P\{f(\boldsymbol{x}_n) \geq 0 \text{ for all } n\} \geq 1 - \delta$.*

*Proof.* By construction of the sequence $\{\beta_n\}$ we know that $P\{f(\boldsymbol{x}) \geq \mu_n(\boldsymbol{x}) - \beta_n \sigma_n(\boldsymbol{x}) \text{ for all } n, \text{ for all } \boldsymbol{x}\} \geq 1 - \delta$. The claim then follows by recalling that the acquisition (6) only selects parameters within $S_n$ and that, by construction of of the safe set, $\mu_n(\boldsymbol{x}) - \beta_n \sigma_n(\boldsymbol{x}) \geq 0$ for all $\boldsymbol{x} \in S_n \setminus \{\boldsymbol{x}_0\}$, in addition to the fact that $f(\boldsymbol{x}_0) > 0$ by assumption. $\qquad\square$

Finally, in Section 4, we claimed that, under the assumptions of the theory, the posterior mean $\mu_n(\boldsymbol{x})$ is bounded by $2\beta_n$ with high probability. The following lemma makes this statement precise. This result formalizes the intuition that for a regular enough GP, in order to get an exploding posterior mean, one needs to be consistently unlucky with the measurement noise.

**Lemma 11.** *Let $f$ be a real valued function on $\mathcal{X}$ and let $\mu_n$ and $\sigma_n$ be the posterior mean and standard deviation of a $GP(\mu_0, k)$ such that it exists a non-decreasing sequence of positive numbers $\{\beta_n\}$ for which $P\big\{f(\boldsymbol{x}) \in [\mu_n(\boldsymbol{x}) \pm \beta_n \sigma_n(\boldsymbol{x})] \, \forall \boldsymbol{x}, \, \forall n\big\} \geq 1 - \delta$. Moreover, assume that $\mu_0(\boldsymbol{x}) = 0$ for all $\boldsymbol{x}$ and that $k(\boldsymbol{x}, \boldsymbol{x}') \leq 1$ for all $\boldsymbol{x}, \boldsymbol{x}' \in \mathcal{X}$. Then it follows that $|\mu_n(\boldsymbol{x})| \leq 2\beta_n$ with probability of at least $1 - \delta$ jointly for all $\boldsymbol{x}$ and for all $n$.*

*Proof.* From the hypothesis, it follows that the following two conditions hold for all $\boldsymbol{x}$ and for all $n$ with probability of at least $1 - \delta$:

$$|f(\boldsymbol{x})| \in \big[0, \beta_0 \sigma_0(\boldsymbol{x})\big] \tag{27}$$

$$\mu_n(\boldsymbol{x}) \in \big[-|f(\boldsymbol{x})| - \beta_n \sigma_n(\boldsymbol{x}), |f(\boldsymbol{x})| + \beta_n \sigma(\boldsymbol{x})\big] \tag{28}$$

From these two conditions, it follows that $|\mu_n(\boldsymbol{x})| \leq \beta_0 \sigma_0(\boldsymbol{x}) + \beta_n \sigma_n(\boldsymbol{x})$ with probability of at least $1 - \delta$. Now, we recall that the sequence $\{\beta_n\}$ is non decreasing by assumption and that the sequence $\{\sigma_n(\boldsymbol{x})\}$ is non increasing by the properties of a GP, which allows us to conclude that $|\mu_n(\boldsymbol{x})| \leq 2\beta_n \sigma_0(\boldsymbol{x})$, which concludes the proof once we recall the assumption that $k(\boldsymbol{x}, \boldsymbol{x}') \leq 1$ for all $\boldsymbol{x}, \boldsymbol{x}' \in \mathcal{X}$. The result can easily be extended to the case of non zero prior mean, by just adding the prior mean as offset in the found upper bound for the posterior mean. $\qquad\square$

## B  Entropy of $\Psi(\boldsymbol{x})$ approximation

In order to analytically compute the mutual information $\hat{I}_n\big(\{\boldsymbol{x}, y\}; \Psi(\boldsymbol{z})\big) = H_n\big[\Psi(\boldsymbol{z})\big] - \mathbb{E}_y\Big[H_{n+1}\big[\Psi(\boldsymbol{z})\big|\{\boldsymbol{x}, y\}\big]\Big]$, we have approximated the entropy of the variable $\Psi(\boldsymbol{x})$ at iteration $n$ with $\hat{H}_n\big[\Psi(\boldsymbol{x})\big]$, given by (4), which we have then used to derive the results presented in the paper. The approximation allowed us to derive a closed expression for the average of the entropy at parameter $\boldsymbol{z}$ after an evaluation at $\boldsymbol{x}$, $\mathbb{E}_y\Big[\hat{H}_{n+1}\big[\Psi(\boldsymbol{z})\big|\{\boldsymbol{x}, y\}\big]\Big]$. This approximation was obtained by noticing that the exact entropy (3) has a zero mean Gaussian shape, when plotted as function of $\mu_n(\boldsymbol{x})/\sigma_n(\boldsymbol{x})$, and then by expanding both the exact expression (3) and a zero mean unnormalized Gaussian in $\mu_n(\boldsymbol{x})/\sigma_n(\boldsymbol{x})$ in their Taylor series around zero. At the second order we obtain, respectively,

$$H_n\big[\Psi(\boldsymbol{x})\big] = \ln(2) - \frac{1}{\pi}\left(\frac{\mu_n(\boldsymbol{x})}{\sigma_n(\boldsymbol{x})}\right)^2 + o\!\left(\left(\frac{\mu_n(\boldsymbol{x})}{\sigma_n(\boldsymbol{x})}\right)^2\right) \tag{29}$$

and

$$c_0 \exp\left\{ -\frac{1}{2\sigma^2}\left(\frac{\mu_n(\boldsymbol{x})}{\sigma_n(\boldsymbol{x})}\right)^2 \right\} = c_0 - c_0\frac{1}{2\sigma^2}\left(\frac{\mu_n(\boldsymbol{x})}{\sigma_n(\boldsymbol{x})}\right)^2 + o\left(\left(\frac{\mu_n(\boldsymbol{x})}{\sigma_n(\boldsymbol{x})}\right)^2\right). \qquad (30)$$

By equating the terms in (29) with the ones in (30), we find $c_0 = \ln(2)$ and $\sigma^2 = \ln(2)\pi/2$, which leads to the approximation for $H_n\big[\Psi(\boldsymbol{x})\big]$ (4) used in the paper: $H_n\big[\Psi(\boldsymbol{x})\big] \approx \ln(2)\exp\left\{ -\frac{1}{\pi\ln(2)}\left(\frac{\mu_n(\boldsymbol{x})}{\sigma_n(\boldsymbol{x})}\right)^2 \right\}$. In Figure 8a we plot (3) and (4) against each other as a function of the mean-standard deviation ratio, while Figure 8b shows the difference between the two. From these two plots, one can see almost perfect agreement between the two functions, with a non negligible difference limited to two small neighborhoods of the $\mu/\sigma$ space.

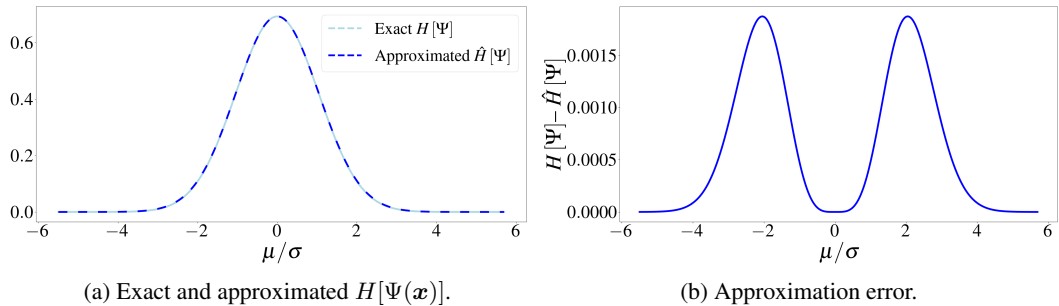

(a) Exact and approximated $H[\Psi(\boldsymbol{x})]$.    (b) Approximation error.

Figure 8: Comparison between exact entropy of $\Psi(\boldsymbol{x})$ (3) and approximated form (4): (a) the two entropies plotted against each other; (b) the approximation error expressed as difference of the two.

## C  Experiments Details

In this Appendix, we collect details about the experiment presented in Section 6. Code for the used acquisition functions can be found at `https://github.com/boschresearch/information-theoretic-safe-exploration`.

ISE selects the next parameter to evaluate according to (6), which is a non convex optimization problem constrained in one of the variables. We find the solution to this problem via constrained gradient ascent with multiple restarts.

**GP samples** For the results shown in Figure 3a we run both ISE and the expansion stage of STAGEOPT on 100 samples from a GP defined on the square $[-2.5, 2.5] \times [-2.5, 2.5]$, with RBF kernel with the following hyperparameters: $\mu_0 \equiv 0$; kernel lengthscale = 0.1; kernel outputscale = 150; $\sigma_\nu^2 = 0.05$, while the safe seed $\boldsymbol{x}_0$ was chosen as the origin: $\boldsymbol{x}_0 = (0, 0)$. For the STAGEOPT runs, we used the code by Berkenkamp et al. (2021), who open-sourced it on GitHub under the MIT license (`https://github.com/befelix/SafeOpt`). As STAGEOPT requires a discretized domain, we used the same uniform discretization of 700 points per dimension for all GP samples. Finally, the percentage of the domain classified as safe is estimated via Monte Carlo sampling. Concerning the safety violations summarized in Table 1, the fact that they are comparable is expected, since in our experiments they all use the posterior GP confidence intervals to define the safe set.

Table 1: Average percentage of safety violations per run over the 100 runs used to obtain Figure 3a.

|  | ISE | SO L=0 | SO L=1 | SO L=5 | SO L=10 |
|---|---|---|---|---|---|
| % of safety violations | $0.04 \pm 0.20$ | $0.01 \pm 0.12$ | $0.03 \pm 0.19$ | $0.05 \pm 0.22$ | $0.05 \pm 0.25$ |

To evaluate whether or not ISE converges to the same safe set as STAGEOPT-like exploration does, we performed the same experiment as for Figure 3a, but with the difference that for this experiment we used a bigger kernel lengthscale of 1.6. For each GP sample, the true reachable safe set is obtained by sampling according to the rule $\boldsymbol{x}_{n+1} \in \arg\max_{S_n} \sigma_n^2(\boldsymbol{x})$, starting from $\boldsymbol{x}_0$, until the uncertainty over the safe set $S_n$ was reduced under the noise variance. We show the results in Figure 9, which

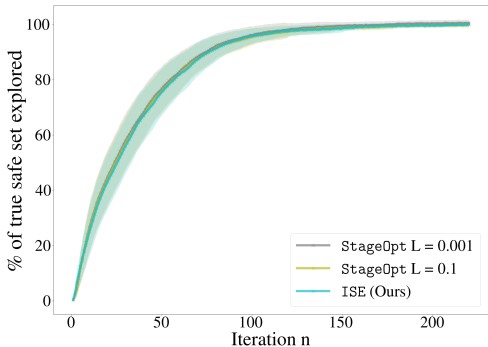

Figure 9: The percentage of the maximally reachable safe set classified as safe is plotted as function of $n$: we can see that ISE eventually leads to discover the same reachable safe set discovered by STAGEOPT-like exploration.

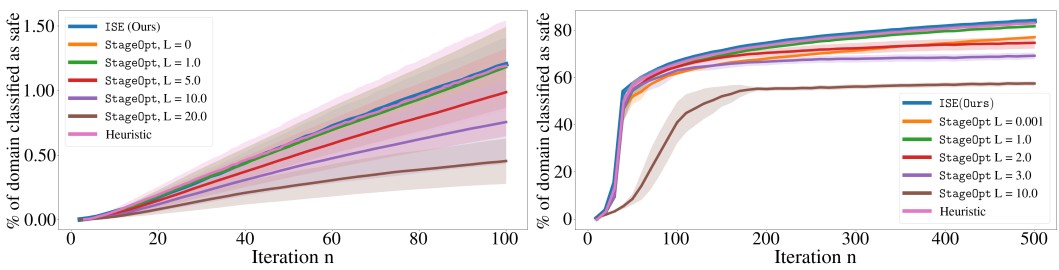

(a) Comparison with STAGEOPT for GP samples.

(b) Comparison in 1D exponential example.

Figure 10: In these plots we show the same results as in Figure 3, but here we plot the standard deviation instead of the standard error.

shows that, indeed, both ISE and STAGEOPT-like exploration lead to the discovery of the same largest safe set. Similarly as for Figure 3a, the percentages that we show are then obtained via Monte Carlo sampling. For the plot in Figure 3b, the constraint function was $f(x) = e^{-x} + 0.05$ and we used a RBF kernel with hyperparameters: $\mu_0 \equiv 0$; kernel lengthscale = 1.2; kernel outputscale = 100; $\sigma_\nu^2 = 0.05$, while the safe seed was $x_0 = 0$, and the domain for the STAGEOPT exploration was composed of 500 points. The chosen function is a slightly offset exponential. On one side of the domain this constraint function becomes increasingly close to the safety threshold, making it hard to explore with high Lipschitz constant. On the other hand, if the Lipschitz constant is too small, the algorithm will prefer to reduce uncertainty away from the border. On the contrary, ISE will always tend to select parameters close to the boundary. This intuition is what justifies the results shown in Figure 3b. In Figure 10 we report the same plots as in Figure 3, but with error bars representing the standard deviation instead of the standard error.

**OpenAI Gym Control**    For the inverted pendulum and cart pole experiments, we used the environments provided by the OpenAI gym (Brockman et al., 2016) under the MIT license. The cart pole environment by default accepts only discrete actions $u_t \in \{0, 1\}$, causing a push of fixed strength either to the left or right. Instead of mapping the output of our linear controller $u_t = \alpha_1 \theta_t + \alpha_2 \dot{\theta}_t + \alpha_3 \dot{s}_t$ to $\{0, 1\}$, we modified the environment to accept continuous actions, corresponding to pushes of varying intensity in the direction specified by the action's sign. For the inverted pendulum experiment, the threshold angular velocity $\dot{\theta}_M$ was set to $0.5$ rad/$s$, with an episode length of 400 steps, and Figure 4 shows one run of ISE using a GP with RBF kernel with the following hyperparameters: $\mu_0 \equiv 0$; kernel lengthscale = 1.3; kernel outputscale = 6.6; $\sigma_\nu^2 = 0.04$. For the cart pole one, the episode length was set to 200 steps and the threshold angle $\theta_M$ was of $0.28$ radians. The GP we used in this case had a RBF kernels with hyperparameters: $\mu_0 \equiv 0$; kernel lengthscale = 0.8; kernel outputscale = 5; $\sigma_\nu^2 = 0.05$. The safe seed $\boldsymbol{\alpha}_0$ was set to $\boldsymbol{\alpha}_0 = (-0.0073, -1.39, 2.01)$, while the domain was set to $[-2, 0] \times [-2, 1.5] \times [-2, 7]$. The average percentage of true safe set classified as safe plotted in Figure 5a is over 100 runs and is estimated via Monte Carlo sampling. For what concerns the comparison about the number of unsafe evaluations in the cart pole task, the average

percentage of safety violations was of $5.02 \pm 0.95$ for the STAGEOPT runs, while for ISE it was of $5.5 \pm 0.98$.

**High dimensional domains**    For the five dimensional experiment we used the same custom LINEBO wrapper for both the ISE and STAGEOPT acquisitions, which at each iteration randomly selects multiple one-dimensional subspaces and then finds the optimum of the respective acquisition function restricted to these subspaces. In these experiments we used a GP with RBF kernel with hyperparameters: $\mu_0 \equiv 0$; kernel lengthscale = 1.6; kernel outputscale = 1; while the safe seed $x_0$ was set to $x_0 = (-0.2)^d$ and the observation noise to $\sigma_\nu^2 = 0.5$.

**Heteroskedastic noise domains**    In these experiments we used the same LINEBO wrapper as in the five dimensional experiment. The GP had a RBF kernel with hyperparameters: $\mu_0 \equiv 0$; kernel lengthscale = 1.6; kernel outputscale = 1; while the safe seed $x_0$ was set to the origin. For what concerns the observation noise, as explained in Section 6, we used heteroskedastic noise, with two different values of the noise variance in the two symmetric halves of the domain. In particular, given a parameter $x = (x_1, x_2, \ldots, x_n)$, we set the noise variance to $\sigma_\nu^2 = 0.05$ if $x_0 \geq 0$, otherwise we set it to $\sigma_\nu^2 = 0.5$. As explained in Section 6, the constraint function is $f(x) = \frac{1}{2}e^{-x^2} + e^{-(x \pm x_1)^2} + 3e^{-(x \pm x_2)^2} + 0.2$, with $x_1$ and $x_2$ given by: $x_1 = (2.7, 0, \ldots, 0)$ and $x_2 = (6, 0, \ldots, 0)$.

**Computational resources**    The experiments were run on a HPC cluster, with each experiment using four Intel Xeon Gold CPUs. All experiments (including early evaluations) amounted to a total of 77020 hours. The Bosch Group is carbon neutral. Administration, manufacturing and research activities do no longer leave a carbon footprint. This also includes GPU clusters on which the experiments have been performed.