# OpenReview forum: "Information-Theoretic Safe Exploration with Gaussian Processes"
_NeurIPS.cc/2022/Conference — NeurIPS 2022 Accept_

### Official Review · Reviewer_HY4V · 2022-06-27

**Rating:** 6
**Confidence:** 3
**Soundness:** 3 good
**Presentation:** 3 good
**Contribution:** 3 good

**Summary:**

This paper provides a newer approach to explore and uncover the safe set of parameters. The exploration is principled and is guaranteed to be safe with high probability. The developments in the theoretical section provide closed-form approximations for the information-gain relevant quantities. The experiments show that their approach can provide the maximally safe set faster than existing methods. The novelty with respect to prior work is that the information gain criteria is used for bayesian optimization with an emphasis on safety. As an illustration of the algorithm, the GP prior is known beforehand with at least one point in the safe set. The next point is chosen to be in the region that is known to be safe such that decreasing the variance around this region will provide maximal information about the safety of any point beyond the safe set. Rather than the Lipschitz constant, the use of a smoothness prior does seem interesting and novel.

**Questions:**

From a theory perspective, [A] proves a finite-time regret bound while this paper only proves an asymptotic guarantee. The first type of bounds are stronger and provide more information about the rate of convergence. In practice, the method actually has an advantage but this is lost in the theory. What is proven here is that once we have found the maximal safe set, the posterior variances of the points inside the safe set will eventually become small. The exploration for uncovering the safe set is optimal at least assuming the prior is correct. Is it relevant to prove the number of samples it would take to uncover the safe set or let us say the regret wrt an algorithm that knows the optimal prior?

The comparison to previous related work is at a very high-level. It would help to provide a deeper theoretical comparison to the 2-3 most related papers. For example, can you place the theorem being proven in the context of the literature in this topic?

[A] Max-value Entropy Search for Efficient Bayesian Optimization - Wang and Jegelka

**Limitations:**

There is a great discussion on limitations of the work.

**Strengths And Weaknesses:**

I believe this is overall a good paper as it is a principled approach to Bayesian optimization and safety. It is improving the Lipschitz constant based methods for sampling-based safety verification.

---

> ### Author Response · Authors · 2022-08-02
> **Response to official review by reviewer HY4V**
>
> Thank you very much for reviewing our paper and for your interesting questions. Here are our responses.
>
> > From a theory perspective, Wang and Jegelka (2017) proves a finite-time regret bound while this paper only proves an asymptotic guarantee. The first type of bounds are stronger and provide more information about the rate of convergence.
> > In practice, the method actually has an advantage but this is lost in the theory. What is proven here is that once we have found the maximal safe set, the posterior variances of the points inside the safe set will eventually become small.
> > The exploration for uncovering the safe set is optimal at least assuming the prior is correct. Is it relevant to prove the number of samples it would take to uncover the safe set or let us say the regret wrt an algorithm that knows the optimal prior?
>
> Our bound in Theorem 1 holds for the exact acquisition function of the method, Eq. (6), that we also use in practical implementation, while in practice Wang and Jegelka (2017) use approximations to sample the function optimum y* required by their theory.
> Aside from that, you are right about Theorem 1: it gives a bound of the number of iterations needed to learn up to a certain confidence, provided that the safe set does not expand.
> Looking into a similar bound for the number of iterations needed to reach the maximally discoverable safe set is certainly a very interesting question, and we will investigate that.
>
> > The comparison to previous related work is at a very high-level. It would help to provide a deeper theoretical comparison to the 2-3 most related papers.
> > For example, can you place the theorem being proven in the context of the literature in this topic?
>
> Theorem 1 proves a statement that is in spirit the key lemma in the SafeOpt theory, i.e. that a non expanding safe set implies a vanishing uncertainty over the sets of interest.
> To prove a bound on the number of iterations needed to identify the safe optimum up to a given confidence, SafeOpt leverages their discrete domain assumption and the explicit dependency of the safe set on the Lipschitz constant.

---

> > ### Comment · Reviewer_HY4V · 2022-08-05
> > **Additional comments and questions**
> >
> > Thank you for the response!
> >
> > It is true that the Lipschitz constant is necessary to prove expansion of safe set. Here, you are leveraging the GP prior information and approximating the mutual information to select most informative points for safe-set expansion. There is an inherent smoothness in the GP prior which might imply a non-zero positive Lipschitz constant. Even though the algorithm need not use this implicit Lipschitz constant, the covariance of the GP prior is the reason why the safe set expands in practice. Can you comment on this?
> >
> > Even in StageOpt and [1], the Lipschitz constant is needed for theoretical purposes and the safe set expansion is performed in experiments using a proxy which uses the mean and variance of the GP posterior. Have you compared with this proxy?
> >
> > [1] Turchetta, Matteo, Felix Berkenkamp, and Andreas Krause. "Safe exploration in finite markov decision processes with gaussian processes." Advances in Neural Information Processing Systems 29 (2016).

---

> > > ### Author Response · Authors · 2022-08-08
> > > **RE: Additional comments and questions**
> > >
> > > Thank you for your questions.
> > >
> > > > It is true that the Lipschitz constant is necessary to prove expansion of safe set. Here, you are leveraging the GP
> > > > prior information and approximating the mutual information to select most informative points for safe-set expansion.
> > > > There is an inherent smoothness in the GP prior which might imply a non-zero positive Lipschitz constant. Even though
> > > > the algorithm need not use this implicit Lipschitz constant, the covariance of the GP prior is the reason why the
> > > > safe set expands in practice. Can you comment on this?
> > >
> > > You are absolutely right that the GP prior encodes a degree of smoothness for the constraint function `f`, similarly
> > > to what the Lipschitz constant does. This covariance is what the GP exploits to generalize (and thus expand) beyond
> > > the current safe set. Empirically this works, but with observation noise it is difficult to analyze theoretically how
> > > exactly the GP is going to generalize. A particular key challenge is also that it is possible for the safe set to
> > > shrink due to an unlucky noise realization. That is why `SafeOpt` and `StageOpt` intersect GP confidence intervals
> > > across iterations and use the Lipschitz constant to generalize. Extending this analysis to instead use the GP itself
> > > is an interesting direction for future work.
> > >
> > > > Even in StageOpt and [1], the Lipschitz constant is needed for theoretical purposes and the safe set
> > > > expansion is performed in experiments using a proxy which uses the mean and variance of the GP posterior. Have you
> > > > compared with this proxy?
> > >
> > >  Indeed, in our experiments the safe set is defined in the same way for both `ISE` and `StageOpt` evaluations.
> > >  Namely, we use only the posterior mean and standard deviation to classify the safe points, as given by equation (2) in
> > >  the paper. If we are not misunderstanding your question, this is precisely the proxy you are referring to.

---

### Official Review · Reviewer_Se3u · 2022-07-08

**Rating:** 6
**Confidence:** 4
**Soundness:** 3 good
**Presentation:** 2 fair
**Contribution:** 3 good

**Summary:**

 The paper focuses on an important stage of GP safe exploration algorithms: safely expanding the size of the set of parameters believed to be safe, in a data-efficient manner.
To do this, the authors propose an information-theoretic approach inspired by other non-safety-constrained Bayesian Optimization (BO) literature.
Compared to standard SafeOpt (Sui et al. "Safe exploration for optimization with Gaussian processes." PMLR15), the method is shown to be more efficient at expanding the safe set.


**Questions:**

why didn't the experimental section analyse the safety spec?

In figure 2 (c), the left-most green evaluation cross is clearly under the orange safety bound line, which suggests that the agent has broken the safety specification in this run. Is this a formatting error or representative of an actual failed run?

Please clarify the point about Theorem 1 above.

**Limitations:**

Limitations on scalability are mentioned in text but not analysed empirically. Negative societal impacts are addressed with a boilerplate sentence in he end of the paper, bit unsatisfactory; probably better just not saying anything.


**Strengths And Weaknesses:**

Strengths:

The originality and contribution the paper makes is good:

* It adapts mutual-information-based methods to the safe set expansion setting, which no other work (that I was able to find) does. Usually information theoretic measures for BO simplify to just sampling at the parameters with maximum variance (Schillinger et al., 2017) or highest UCB, but in this case it is more interesting and the authors are able to come up with a function to optimise to find the maximum information gain about expanding the safe set.

* Their acquisition function simplification to make optimization tractable seems justified and well explained in the appendix.
This is not the only work to avoid the inclusion of a Lipschitz constant hyperparameter (Berkenkamp et al. "Safe controller optimization for quadrotors with Gaussian processes.", ICRA16) (they use hypothesised GP observations for safe set expansion) or to avoid SafeOpt-style discretisation (Schillinger et al., 2017) (this is another method that works in continuous parameter space). However, these two problems are solved here in an elegant way.
In particular, it avoids the basic heuristic/arbitrariness of the expander state definition in SafeOpt.

* I believe the authors have covered all the relevant related work – although not sure the paper experimentally compares to enough related work (see below).

* I believe the clarity of the paper is generally good until the experiments section.


Weaknesses:

* The acquisition function simplification should be more explained in the main body of the paper. Furthermore, it would be good to have further discussion on the approximation on the rest of the paper, e.g. Theorem 1.

* The evaluation with GP samples, shown in Figure 3, is confusing. The only alternative method comparison is standard StageOpt with differing Lipschitz (L) parameter values. The justification for the choice of plotted L-values isn’t clear, and furthermore there is a clear trend of decreasing L leading to better performance. From this graph, it would suggest that reducing L further <1 might result in it outperforming ISE.
For those with background knowledge, it is clear why it’s not possible to keep reducing L. Eventually, reducing L enough will result in StageOpt classifying unsafe states as safe and then failing the safety specification. However this section contains no discussion of whether/how any algorithms fail the safety specification, and only shows the first few % of safe set exploration. Evaluation should include plots on how many time s the safety spec is broken, as that is a crucial part of the problem

* I think evaluation might have benefited comparing to SafeOpt variants that do not require the Lipschitz constant, such as (Berkenkamp et al. "Safe controller optimization for quadrotors with Gaussian processes.", ICRA16). That would better differentiate which performance improvements are from more intelligent expansion behaviour, vs tuning of the Lipschitz constant.
Figure 6 does not have enough baseline comparison. Only one StageOpt plot is shown in 6 (a) and it is not clear which Lipschitz value was used for this. There is no StageOpt plot in 6 (b) and it is unclear why. It's also unclear why the plot only shows the first 5% of exploration.

* There should be some experiments on scalability, e.g. it would be nice to see a scalability plot on number of dimensions.



Other  points:

The last paragraph of related work should also mention Turchetta, Matteo, Felix Berkenkamp, and Andreas Krause. "Safe exploration in finite markov decision processes with gaussian processes." Advances in Neural Information Processing Systems 29 (2016).

“This process is made more efficient by SafeOpt, which restricts exploration to parameters that are close to the boundary of the current set of safe parameters (Sui et al., 2015)”: I think it is more accurate to say that StageOpt does this restriction (in its expansion phase). SafeOpt can also choose to sample potential maximisers in M_t, not just from the expander set G_t.

Clarity-wise, I don’t fully understand how Theorem 1 justifies the sentence “in practice it will first focus on reducing the uncertainty in areas of the safe set that are most informative about parameters whose classification is still uncertain (e.g. areas close to the boundary of the safe set), and only eventually turns to learning about the inside of the safe set”. If it doesn’t justify that sentence, the theorem doesn’t seem to actually prove anything about safe set expansion.

---

> ### Author Response · Authors · 2022-08-02
> **Response to official review by reviewer Se3u - Part 1**
>
> Thank you for reviewing our paper and for your insightful suggestions on how to improve it, especially related to the experiments section. In the following are the details on how we have addressed your comments. The response is divided into two comments (Part 1 and Part 2), as OpenReview limits the number of allowed characters per comment.
>
> > The acquisition function simplification should be more explained in the main body of the paper.
>
> We obtain the approximation (4) for the entropy as the Taylor expansion of the true entropy and a generic Gaussian up to the second order. We have now added an additional reference to this in the main body of the paper.
> For such a Taylor expansion there are well-known upper bounds on the approximation error, and Fig. 8 shows that numerical errors between the approximated entropy and the true entropy are negligible (Fig. 8).
> We use the approximated entropy (4) to define the acquisition function, and all the following analysis (including the theoretical results and Theorem 1) is on that expression.
> To clarify this point, we have now changed the notation in the manuscript to reflect that the analysis and the proofs are with respect to the approximated entropy.
>
> > The evaluation with GP samples, shown in Figure 3, is confusing, it would suggest that reducing L further <1 might result in it outperforming ISE.
>
> Thank you for the observation, this is a good point.
> We have now added to Fig. 3a the results for L=0 and, in Fig. 3b, the results of another experiment that shows how in certain scenarios both too high and too low Lipschitz constant lead to poor performance.
>
> > Evaluation should include plots on how many time s the safety spec is broken, as that is a crucial part of the problem.
>
> This is also a valid point.
> We had originally decided not to include those since safety is purely determined by the GP safety intervals, which are shared across all methods in the experiment section.
> We have now added a table in the Appendix C showcasing the safety violation information corresponding to Fig. 3a.
>
> > I think evaluation might have benefited comparing to SafeOpt variants that do not require the Lipschitz constant.
>
> This is a good suggestion and, in fact, the  StageOpt implementation that we used does partially take inspiration from the mentioned paper, as the safe set is not computed using the Lipschitz constant, but the GP confidence intervals.
> We now explicitly state this in the manuscript. We will also include a direct comparison to Berkenkamp et al. In the final paper, although their approach does not yield any theoretical guarantees on learning the safe set.
>
> > Figure 6 does not have enough baseline comparison. It's also unclear why the plot only shows the first 5% of exploration.
>
> Your comments about Fig. 6 are completely justified.
> Concerning the label of the Y axis of both plots in Fig. 6, they were just the wrong ones, and we apologize for having overseen that.
> The quantity plotted was the found optimum and not the percentage of domain classified as safe. We have corrected the manuscript accordingly.
> We have also added more StageOpt baselines for more dimensions and considerations about safety.
>
> > There should be some experiments on scalability, e.g. it would be nice to see a scalability plot on number of dimensions.
>
> As pointed out in Section 5, although not suffering of the curse of dimensionality due to the domain discretization, our method still must handle a 2d-dimensional non-convex optimization problem, that becomes increasingly challenging with the number of dimensions.
> Therefore, for high dimensional problems, we combine our method with the random projections suggested by the LineBO algorithm, in the same way we do with StageOpt in high dimensions.

---

> ### Author Response · Authors · 2022-08-02
> **Response to official review by reviewer Se3u - Part 2**
>
> And here are the responses to the rest of your comments.
>
> > The last paragraph of related work should also mention Turchetta, Matteo, Felix Berkenkamp, and Andreas Krause. "Safe exploration in finite markov decision processes with gaussian processes."
>
> We have added the reference.
>
> > “This process is made more efficient by SafeOpt, which restricts exploration to parameters that are close to the boundary of the current set of safe parameters (Sui et al., 2015)”:
> > I think it is more accurate to say that StageOpt does this restriction (in its expansion phase). SafeOpt can also choose to sample potential maximizers in M_t, not just from the expander set G_t.
>
> We apologize for the misleading phrasing.
> Implicitly we were referring to the exploration component of SafeOpt focused on expanding the safe set (i.e. the definition of the set of expanders G_t), but is true that we can make this more explicit.
> We have now modified the manuscript accordingly.
>
> > Clarity-wise, I don’t fully understand how Theorem 1 justifies the sentence
> > “in practice it will first focus on reducing the uncertainty in areas of the safe set that are most informative about parameters whose classification is still uncertain (e.g. areas close to the boundary of the safe set),
> > and only eventually turns to learning about the inside of the safe set”.
>
> Thanks for highlighting the lack of clarity on this point.
> In practice we observed that initially the acquisition function focuses on the boundary of the safe set, since those parameters are the most informative about the safety of parameters outside of that set.
> Once it has reduced the uncertainty on the boundary, the acquisition will then try to squeeze out all the information from the rest of the safe set, and this is what Theorem 1 guarantees.
> This result is also linked with exploration because it rules out the possibility that the proposed acquisition function forever leaves the uncertainty high in areas of the safe set that, if better understood, would lead to an expansion of the safe set.
> We have updated the manuscript making this point clearer.
>
> > In figure 2 (c), the left-most green evaluation cross is clearly under the orange safety bound line
>
> Thank you for spotting this unfortunate illustration flaw. There indeed was a safety violation in the figure due to an unlikely noise realization.
> We replaced the plots with new ones obtained with a different constraint function that does not contain any unsafe evaluation.
> However, it is worth mentioning that the theory guarantees safety only with high probability, and that the safety is with respect to the true value of the constraint function, and not the noisy evaluation, which is what we show in the plot.

---

> > ### Comment · Reviewer_Se3u · 2022-08-04
> > **quick questions**
> >
> > 1. Figure 3(b) doesn't seem to have error bars when it says that it does?
> >
> > 2. Could you discuss the complexity of the optimisation step, in particular how much slower it is than  StageOpt's next sample point choice step?

---

> > > ### Author Response · Authors · 2022-08-08
> > > **RE: quick questions**
> > >
> > > Thank you for your questions.
> > >
> > > > Figure 3(b) doesn't seem to have error bars when it says that it does?
> > >
> > > Similarly to Fig 3a, also in Fig. 3b the error bars correspond to the standard error of the mean, which is very small.
> > > This is a desired feature, as it indicates that the number of run experiments was sufficient to obtain conclusive
> > > results about the average performance. We will include plots with the standard deviation in the appendix
> > >
> > > > Could you discuss the complexity of the optimisation step, in particular how much slower it is than StageOpt's
> > > > next sample point choice step?
> > >
> > > Concerning the complexity of the optimization step, in theory both StageOpt and ISE deal with a costrained 2D
> > > acquisition function that they need to optimize to find $x_{n+1}$. For StageOpt, if we call $x$ the variable that is
> > > constrained to the safe set, ad $z$ the variable that is bound to be outside of the safe set, the acquisition function
> > > would be of the form $\sigma_n(x) \mathcal{I}(\mu_n(x) + \beta \sigma_n(x) - Ld(x, z) > 0)$, where $\mu_n(x)$ and $\sigma_n(x)$ are
> > > the posterior mean and standard deviation, and $\mathcal{I}( \cdot )$ is the indicator function. For ISE the acquisition is just
> > > the approximated mutual information $I(y(x); \Psi(z))$ derived in the paper. The average evaluation cost of the two
> > > acquisition functions (without optimization) on a two-dimensional domain over 50 evaluations are comparable:
> > >
> > > - StageOpt: 3.5 +- 0.6 milliseconds
> > > - ISE: 3.8 +- 0.2 milliseconds
> > >
> > > Comparing the overall optimization is challenging, since it is heavily dependent on implementation details. Moreover,
> > > we are comparing a continuous implementation for ISE, which has to do iterative gradient steps, with an algorithm that
> > > relies on a discretization and evaluations can be done in batch (faster in Python).
> > >
> > > That being said, the average duration of exploring a two-dimensional constraint function over 100 evaluations are:
> > >
> > > - StageOpt: 1979 +- 755 milliseconds
> > > - ISE: 3064 +- 1042 milliseconds
> > >
> > > Please notice that we perform multi-start gradient descent in the ISE optimization step. The number of restarts has not
> > > been optimized to minimize execution time. We will investigate a more high-performance implementation in time for the
> > > camera-ready version of the paper.

---

> > ### Comment · Reviewer_Se3u · 2022-08-09
> > **thanks**
> >
> > Thanks for the clarifications, I raised my score

---

### Official Review · Reviewer_ZWAM · 2022-07-10

**Rating:** 6
**Confidence:** 3
**Soundness:** 3 good
**Presentation:** 3 good
**Contribution:** 3 good

**Summary:**

Optimization of an unknown function under unknown constraints is a problem which has been often addressed based on Gaussian process surrogate models in the past. Due to the unknown constraints, existing approaches typically start with a given set of safe parameters, which is subsequently expanded. In order to ensure the safety of the approach, this expansion must not lead to constraint violations. State-of-the-art methods determine evaluation points for the expansion of the safe set by finding the most uncertain point in discretized sets. However, this leads to additional hyperparameters and tends to scale poorly to high dimensional input domains. This paper aims to avoid the issues resulting from discretization by employing the information gain of the binary random variable describing the constraint violation at a given input point. Since this information gain cannot be computed exactly in practice, an efficient approximation is proposed. It is shown that determining evaluation points using the proposed criterion is guaranteed to achieve any upper bounds on the posterior GP variance in a finite number of evaluation points if the safe set is not expanded anymore. The practical strengths of the method are demonstrated in simulations with GP sample functions and control examples. In particular in high dimensional problems, a significantly better exploration of the true safe region can be observed.

**Questions:**

Apart from my comments mentioned above, I only have minor suggestions/questions:
- Highlighting ISE using a bold line in Figs. 3 and 6 might be a good idea
- How is beta chosen in the simulations?
- Line 47: in $\rightarrow$ on
- Lines 231-232: exploration exploration
- Lines 262-263: the true safe for this problem
- The y axis label of Fig. 6 seems wrong. I think it should be the found optimum.
- Line 280: I think it should be referred to Fig. 5b instead of 6b here.


**Limitations:**

I think the limitations of the method and potential societal impact are generally well discussed. While the choice of the kernel function is highlighted as a potential challenge, it is not sufficiently discussed that the RKHS bound also needs to be known for the theory to hold. This bound is generally difficult to obtain in practice, which crucially limits practical safety guarantees. In my opinion, this should be discussed in the limitations section.

**Strengths And Weaknesses:**

The paper is on an interesting and relevant topic. The proposed ideas are novel to the best of my knowledge and nicely relate to entropy search and similar approaches in unconstrained Bayesian optimization. The paper combines theoretical analysis and demonstration of practical advantages well. In general, the paper is well-structured and the presentation of the method is excellent in my opinion. Merely in the supplemental material I have noticed that everything is proven for the approximation proposed by the authors, while it is stated that the analysis is performed for the mutual information itself. It took me some time to realize this discrepancy, such that I would recommend to clarify this. The derivation of the theoretical results looks quite straightforward, but the result is certainly interesting and fits well into the overall paper. However, it does not become clear if the scenario analyzed in Theorem 1 does actually ever occur. The reason for this is that it requires that the safe set needs to stop growing at some point in time, which could potentially never happen if the increase of the size of the safe set becomes smaller and smaller every iteration, but never becomes zero. This case is also not discussed in the paper, such that the significance of the theoretical results strongly diminishes in my opinion. I did not check the theoretical results in detail, but they seem to be correct to the best of my knowledge. The simulations are properly executed with sufficiently many random seeds including functions satisfying the assumptions of the approach and functions for which this is not clear. The effect of the dimensionality on the performance of the different approaches is illustrated nicely. I merely have a small technical comment regarding the statement in section 6 that the ISE approach is evaluated on samples from a GP because this would allow to test ISE under the assumptions of the theory. While this is true for the exploration, the usage of GP sample function poses the theoretical challenge that samples from a GP are well-known to almost surely have an unbounded RKHS norm. Therefore, the definition of the safe set based on (Chowdhary and Gopalan, 2017) does not work anymore because it requires a bound for the RKHS norm.

---

> ### Author Response · Authors · 2022-08-02
> **Response to official review by reviewer ZWAM**
>
> Thank you for reviewing our paper and for providing useful comments on how to improve clarity and transparency in some passages. We have addressed your comments as follows:
>
> > everything is proven for the approximation proposed by the authors, while it is stated that the analysis is performed for the mutual information itself. It took me some time to realize this discrepancy, such that I would recommend to clarify this.
>
> You are right, all the proofs and the analysis are for the expressions obtained using the approximation in eq. (4).
> We have updated the manuscript and changed the notation for the approximated mutual information to make it explicit that we reason about the expression obtained via the approximated entropy (4).
>
> > It does not become clear if the scenario analyzed in Theorem 1 does actually ever occur.
>
> You raise a good point. Without additional assumptions, it is possible that the safe set continues to expand forever (e.g., if the true safe set is unbounded).
> However, Theorem 1 ensures that we do not get stuck in some sub-region of the safe set, since whenever the safe set does not expand we learn the constraint function perfectly within the safe set.
> Guaranteeing how frequently this happens is challenging and the only theoretical results in that direction build on the SafeOpt algorithm, which is limited to finite spaces and uses the Lipschitz constant to classify states as safe rather than directly relying on the GP model.
> We updated the manuscript with some considerations in this regard.
>
> > The usage of GP sample function poses the theoretical challenge that samples from a GP are well-known to almost surely have an unbounded RKHS norm.
>
> You are right that a GP sample has unbounded norm in the RKHS space, and we apologize that our phrasing has not been precise on this point.
> We do not use actual functional samples from a GP, but approximate them  by sampling the GP at a finite number of points.
> This construction leads to constraint functions that do have bounded RKHS norm. We added this information to the manuscript.
>
> > Highlighting ISE using a bold line in Figs. 3 and 6 might be a good idea.
>
> Thank you for the suggestion, we have updated the plots accordingly.
>
> > How is beta chosen in the simulations?
>
> We apologize that the text was unclear about the experiment setup. Similarly to previous related work, in the simulations we chose beta as a constant independent of n, with a default value of 2. We updated the manuscript adding this information. Using a constant beta,  safety is no longer guaranteed jointly for all n, but only per iteration. We want to highlight that unlike previous methods our mutual-information criterion is independent of beta. Thus scaling up beta would not significantly impact our method, but StageOpts exploration criterion would consider a significantly larger set of expanders (since it is constructed using the upper bounds `\mu_n + \beta_n\sigma_n`) and would become less data-efficient.
>
> > Various typos:
>
> Thank you very much for highlighting the typos and wrong figure reference. We have now fixed them.
>
> > The y axis label of Fig. 6 seems wrong
>
> The Y labels in Fig. 6 were indeed wrong. We have corrected them.
>
> > It is not sufficiently discussed that the RKHS bound also needs to be known for the theory to hold.
>
> Once again, you are right about the challenge posed by the RKHS norm. We have included a discussion about this in the limitations section.

---

> > ### Comment · Reviewer_ZWAM · 2022-08-06
> > **Re: Response to official review by reviewer ZWAM**
> >
> > Thank you for answering my questions!

---

### Meta-Review · Area_Chair_46EH · 2022-08-26

**Recommendation:** Accept
**Confidence:** Certain

**Metareview:**

I went through the reviews and authors' responses. The reviewers are experienced in this field and provided good quality reviews. Scores are the same: weak accept. I also went through the paper and considered it reached the bar of NeurIPS.

**Award:**

No

---

### Decision · Program_Chairs · 2022-09-14

Accept